# Stable Diffusion is Unstable

**Chengbin Du, Yanxi Li, Zhongwei Qiu, Chang Xu**
School of Computer Science, Faculty of Engineering
University of Sydney, Australia
chdu5632@uni.sydney.edu.au, yali0722@uni.sydney.edu.au
zhongwei.qiu@sydney.edu.au, c.xu@sydney.edu.au

## Abstract

Recently, text-to-image models have been thriving. Despite their powerful generative capacity, our research has uncovered a lack of robustness in this generation process. Specifically, the introduction of small perturbations to the text prompts can result in the blending of primary subjects with other categories or their complete disappearance in the generated images. In this paper, we propose **Auto-attack on Text-to-image Models (ATM)**, a gradient-based approach, to effectively and efficiently generate such perturbations. By learning a Gumbel Softmax distribution, we can make the discrete process of word replacement or extension continuous, thus ensuring the differentiability of the perturbation generation. Once the distribution is learned, ATM can sample multiple attack samples simultaneously. These attack samples can prevent the generative model from generating the desired subjects without tampering with the category keywords in the prompt. ATM has achieved a 91.1% success rate in short-text attacks and an 81.2% success rate in long-text attacks. Further empirical analysis revealed three attack patterns based on: 1) variability in generation speed, 2) similarity of coarse-grained characteristics, and 3) polysemy of words. The code is available at https://github.com/duchengbin8/Stable_Diffusion_is_Unstable

## 1 Introduction

In recent years, the field of text-to-image generation has witnessed remarkable advancements, paving the way for groundbreaking applications in computer vision and creative arts. Notably, many significant developments have captured the attention of researchers and enthusiasts, such as Stable Diffusion [28, 32], DALL·E2 [26, 27, 23] and Midjourney [19]. These developments push text-to-image synthesis boundaries, fostering artistic expression and driving computer vision research.

Despite the remarkable progress in text-to-image models, it is important to acknowledge their current limitations. One significant challenge lies in the instability and inconsistency of the generated outputs. In some cases, it can take multiple attempts to obtain the desired image that accurately represents the given textual input. An additional obstacle revealed by recent researches [33, 2, 8] is that the quality of generated images can be influenced by specific characteristics inherent to text prompts. Tang et al. [33] proposes DAAM, which performs a text-image attribution analysis on conditional text-to-image model and produces pixel-level attribution maps. Their research focuses on the phenomenon of feature entanglement and uncovers that the presence of cohyponyms may degrade the quality of generated images and that descriptive adjectives can attend too broadly across the image. Attend-and-Excite [2] investigates the presence of catastrophic neglect in the Stable diffusion model, where the generative model fails to include one or more of the subjects specified in the input prompt. Additionally, they discover instances where the model fails to accurately associate attributes such as colors with their respective subjects. Although those works have some progress, there is still work to be done to enhance the stability and reliability of text-to-image models, ensuring consistent and satisfactory results for a wide range of text prompts.

One prominent constraint observed in those works related to the stability of text-to-image models lies in their dependence on manually crafted prompts for the purpose of vulnerability identification. This approach presents several challenges. Firstly, it becomes difficult to quantify the success and

37th Conference on Neural Information Processing Systems (NeurIPS 2023).

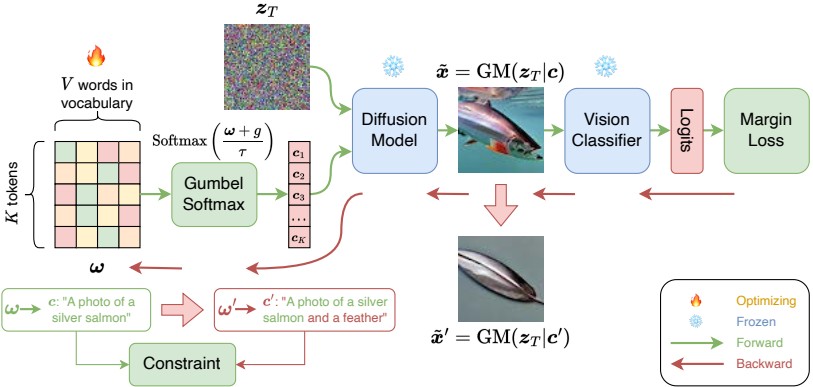

Figure 1: The overall pipeline of our attack method. The selection of words is relaxed to be differentiable by using a Gumbel Softmax with temperature $\tau$. When $\tau \to 0$, the Gumbel Softmax outputs exhibit proximity to one-hot vectors while retaining differentiability. After an image is generated, a CLIP [25] classifier and a margin loss are employed to optimize $\omega$ aiming to generate images that cannot be correctly classified by CLIP.

failure cases accurately, as the evaluation largely depends on subjective judgments and qualitative assessments. Additionally, the manual design of prompts can only uncover a limited number of potential failure cases, leaving many unexplored scenarios. Without a substantial number of cases, it becomes challenging to identify the underlying reasons for failures and effectively address them. To overcome these limitations, there is a growing demand for a learnable method that can automatically identify failure cases, enabling a more comprehensive and data-driven approach to improve text-to-image models. By leveraging such an approach, researchers can gain valuable insights into the shortcomings of current methods and develop more robust and reliable systems for generating images from textual descriptions.

In this paper, we propose **Auto-attack on Text-to-image Models (ATM)**, to automatically and efficiently generate attack prompts with high similarity to given clean prompts (Fig. 1). We use Stable Diffusion [28, 32] as our target model. With the open-source implementation and model parameters, we can generate attack prompts with a white-box attack strategy. Remarkably, those attack prompts can transfer to other generative models, enabling black-box attacks. Two methods to modify a text prompt are considered, including **replacing** an existing word or **extending** with new ones. By incorporating a Gumbel Softmax distribution into the word embedding, the discrete modifications can be transformed into continuous ones, thereby ensuring differentiability. To ensure the similarity between clean and the attack prompts, a binary mask that selectively preserves the noun representing the desired object is applied. Moreover, two constraints are imposed: a **fluency constraint** that ensures the attack prompt is fluent and easy to read, and a **similarity constraint** that regulates the extent of semantic changes.

After the distribution is learned, ATM can sample multiple attack prompts at once. The attack prompts can prevent the diffusion model from generating desired subjects without modifying the nouns of desired subjects and maintain a high degree of similarity with the original prompt. We have achieved a 91.1% success rate in short-text attacks and a 81.2% success rate in long-text attacks. Moreover, drawing upon extensive experiments and empirical analyses employing ATM, we are able to disclose the existence of three distinct attack patterns, each of which corresponds to a vulnerability in the generative model: 1) the variability in generation speed; 2) the similarity of coarse-grained characteristics; 3) the polysemy of words. In the following, we will commence with an analysis of the discovered attack patterns in Section 4, followed by a detailed exposition of our attack method in Section 5.

In addition, ATM can automatically and efficiently generate plenty of successful attack prompts, which serves as a valuable tool for investigating vulnerabilities in text-to-image generation pipelines. This method enables the identification of a wider range of attack patterns, facilitating a comprehensive examination of the underlying causes. It will inspire the research community and garner increased attention toward exploring the vulnerabilities present in contemporary text-to-image models and will foster further research concerning both attack and defensive mechanisms, ultimately leading to enhanced security within the industry.

## 2 Related Work

### 2.1 Diffusion Model.

Recently, the diffusion probabilistic model [30] and its variants [12, 21, 31, 28, 29, 4, 5, 15] have achieved great success in content generation [31, 13, 29], including image generation [12, 31], conditional image generation [28], video generation [13, 36], 3D scenes synthesis [18] and so on. Specifically, DDPM [12] adds noises to images and learns to recover images from noises step by step. Then, DDIM [31] improves the generation speed of the diffusion model by skipping steps inference. Then, the conditional latent diffusion model [28] formulates the image generation in latent space guided by multiple conditions, such as texts, images, and semantic maps, further improving the inference speed and boarding the application of the diffusion model. Stable diffusion [28], a latent text-to-image diffusion model capable of generating photo-realistic images given any text input, and its enhanced versions [37, 14, 20], have been widely used in current AI-generated content products, such as Stability-AI [32], Midjourney [19], DALL·E2 [23], and Runaway [7]. However, these methods and products cannot always generate satisfactory results from the given prompt. Therefore, in this work, we aim to analyze the robustness of stable diffusion in the generation process.

### 2.2 Vulnerabilities in Text-to-image Models.

With the open-source of Stable Diffusion [28], text-to-image generation achieves great process and shows the unparalleled ability on generating diverse and creative images with the guidance of a text prompt. However, there are some vulnerabilities have been discovered in existing works [8, 2, 33]. Typically, StructureDiffusion [8] discovers that some attributes in the prompt are not assigned correctly in the generated images, thus they employ consistency trees or scene graphs to enhance the embedding learning of the prompt. In addition, Attend-and-Excite [2] also introduces that the Stable Diffusion model fails to generate one or more of the subjects from the input prompt and fails to correctly bind attributes to their corresponding subjects. These pieces of evidence demonstrate the vulnerabilities of the current Stable Diffusion model. However, to the best of our knowledge, no work has systematically analyzed the vulnerabilities of the Stable Diffusion model, which is the goal of this work.

## 3 Preliminary

The architecture of the Stable Diffusion [28] comprises of an encoder $\mathcal{E} : \mathcal{X} \rightarrow \mathcal{Z}$ and a decoder $\mathcal{D} : \mathcal{Z} \rightarrow \mathcal{X}$, where $\tilde{x} = \mathcal{D}(\mathcal{E}(x))$. Additionally, a conditional denoising network $\epsilon_\theta$ and a condition encoder $\tau_\theta$ are employed. In the text-to-image task, the condition encoder is a text encoder that maps text prompts into a latent space. The text prompt is typically a sequence of word tokens $c = \{c_1, \ldots, c_K\}$, where $K$ is the sequence length. During the image generation process, a random latent representation $z_T$ is draw from a distribution such as a Gaussian distribution. Then, the reverse diffusion process is used to gradually recover a noise-free latent representation $z$. Specifically, a conditional denoising network $\epsilon_\theta(z_t, t, \tau_\theta(c))$ is trained to gradually denoise $z_t$ at each time step $t = T, \ldots, 1$ to gradually reduce the noise level of $z_t$, where the condition $c$ is mapped in to a latent space using $\tau_\theta(c)$ maps and the cross-attention between the condition and features is incorporated in $\epsilon_\theta$ to introduce the condition. Finally, the denoised latent representation $z$ is decoded by a decoder $\mathcal{D}$ to produce the final output $\tilde{x}$. The aim of our study is to introduce slight perturbations to the text prompts, thereby inducing the intended object to be blended with other objects or entirely omitted from the generated images. For the sake of simplification, in the forthcoming sections, we shall represent the image generation process using the GM as $\tilde{x} = \mathrm{GM}(z_T | c)$.

## 4 Vulnerabilities of Stable Diffusion Model

By applying the attack method proposed in this paper, a variety of attack text prompts can be generated and analyzed. In this section, the identified attack patterns are discussed. Details of the attack method are introduced in Section 5. We have discovered three distinct patterns of impairment in Stable Diffusion model: 1) Variability in Generation Speed, where the model struggles to reconcile the differences in generation speed among various categories effectively. 2) Similarity of Coarse-grained Characteristics, which arises from the feature entanglement of global or partial coarse-grained characteristics, which possess a high degree of entanglement. 3) Polysemy of Words, which involves the addition of semantically complementary words to the original prompt, resulting in the creation of images that contain brand-new content and are not related to the original category.

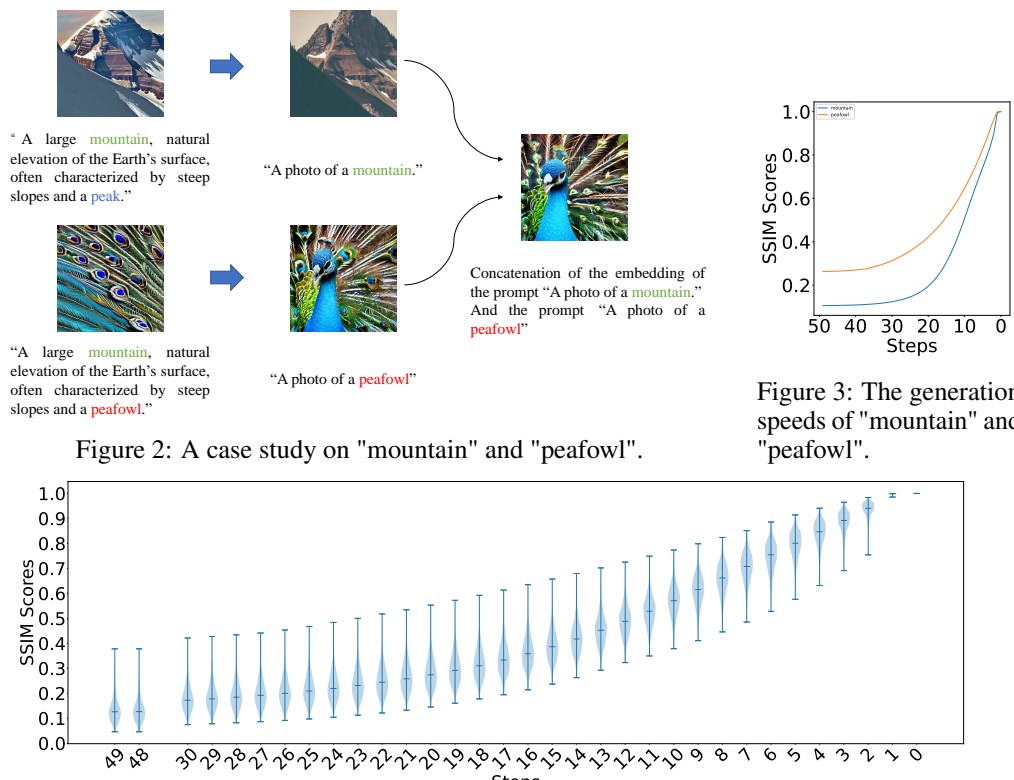

Figure 2: A case study on "mountain" and "peafowl".

Figure 3: The generation speeds of "mountain" and "peafowl".

Figure 4: A violin plot illustrating the generation speeds of $1,000$ images of various classes. The horizontal axis represents the number of steps taken, ranging from 49 to 0, while the vertical axis displays the SSIM scores. The width of each violin represents the number of samples that attained a specific range of SSIM scores at a given step.

## 4.1 Variability in Generation Speed

**Observation 1.** *When a given text prompt contains a noun $A$ representing the object to be generated, the introduction of another noun $B$ into the prompt, either through addition or replacement, leads to the disappearance of noun $A$ from the generated image, replaced instead by noun $B$.*

In Observation 1, we identify a phenomenon when replacing or adding a noun in the description in a text prompt, the new noun will lead to the completely disappear of the desired subject. As shown in Fig. 2, if the noun "peak" is replaced by "peafowl", the desired subject "mountain" disappears and the new subject "peafowl" is generated. To further investigate this phenomenon, we use two short prompts $c_1$ and $c_2$ containing "mountain" and "peafowl", respectively, to exclude the influence of other words in the long prompts. To eliminate the additional impact of all possible extraneous factors, such as contextual relationships, they are embedded separately and then concatenated together: $\mathrm{Concat}(\tau_\theta(c_1), \tau_\theta(c_2))$. The result shows that almost no element of the mountain is visible in the generated image (Fig. 2).

Further analysis reveals a nontrivial difference in the generation speeds of the two subjects. To define the generation speed, a metric to measure the distance from a generated image $\tilde{x}_t$ at a given time step $t = T - 1, \ldots, 0$ to the output image $\tilde{x}_0$ is desired (note, $\tilde{x}_T$ is the initial noise). We use the structural similarity (SSIM) [34] as the distance metrics: $s(t) := \mathrm{SSIM}(\tilde{x}_t, \tilde{x}_0)$. Therefore, the generation speed can be formally defined as the derivative of the SSIM regarding the time step: $v(t) := ds(t)/dt \approx (s(t) - s(t+1))/\Delta t$, where $\Delta t = 1$. Thereby, we propose our Pattern 1.

**Pattern 1** (Variability in Generation Speed). *Comparing the generation speeds ($v_1$ and $v_2$) of two subjects ($S_1$ and $S_2$), it can be observed that the outline of the object in the generated image will be taken by $S_1$ if $v_1 > v_2$. And it can be inferred that $S_2$ will not be visible in the resulting image.*

We further generates $1,000$ images of various classes with the same initial noise and visualize their generation speed in Fig. 4 as a violin plot. The SSIM distance from the generated images at each

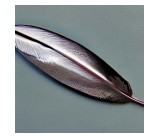 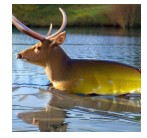 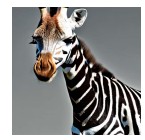 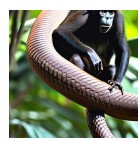

(a) A photo of a silver salmon and a feather.

(b) A photo of a tench and a buck.

(c) A photo of a zebra and a giraffe.

(d) A photo of a howler monkey and a snake.

Figure 5: Images generated by the template "A photo of $A$ and $B$".

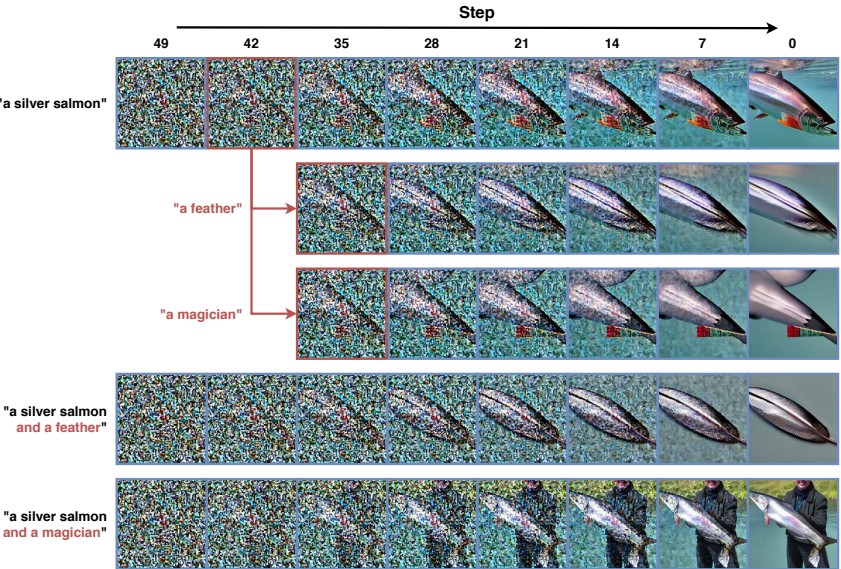

Figure 6: The first row illustrates the generation process with the prompt "a photo of a silver salmon". The second row, based on the forty-second step of the first row, shows the generation process with the prompt "a photo of a feather". The third row, also building upon the forty-second step of the first row, presents the generation procedure when the prompt is "a photo of a magician". The fourth row depicts the generation process in the presence of feature entanglement. The fifth row demonstrates the generation process for two distinct categories without feature entanglement.

step to the final image is calculated. The horizontal axis represents $49 \sim 0$ steps, while the vertical axis represents the SSIM scores. Each violin represents the distribution of the SSIM scores of the $1,000$ images in a step, with the width corresponds to the frequency of images reaches the score. In the early stages of generation, the median of the distribution is positioned closer to the minimum value, indicating that a majority of classes exhibit slow generation speeds. However, the presence of a high maximum value suggests the existence of classes that generate relatively quickly. In the middle stages of generation, the median of the distribution gradually increases, positioning itself between the maximum and minimum values. In the later stages of generation, the median of the distribution is positioned closer to the maximum value, indicating that a majority of classes are nearing completion. However, the persistence of a low minimum value suggests the presence of classes that still exhibit slow generation speeds. This analysis highlights the variation in generation speeds across different classes throughout the entire generation process. This phenomenon can be interpreted as a characteristic of the generation process, where different classes exhibit varying speeds throughout the stages. It is possible that certain classes have inherent complexities or dependencies that cause them to generate more slowly. Conversely, other classes may have simpler structures or fewer dependencies, leading to faster generation.

## 4.2 Similarity of Coarse-grained Characteristics

**Observation 2.** *When a text prompt contains a noun $A$ representing the object to be generated, the introduction of another noun $B$, which describes an object with similar coarse-grained characteristics to the object represented by noun $A$, into the prompt, either through addition or replacement, results in the generation of an image that contains an object combining elements of both nouns $A$ and $B$.*

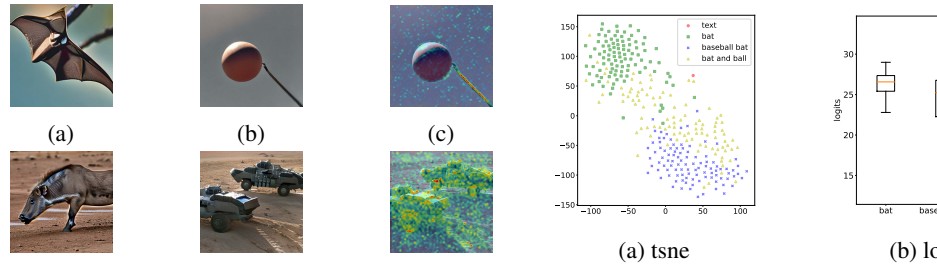

Figure 7: a) "A photo of a bat"; b) "A photo of a bat and a ball;" c) Heat map of the word "bat" in generated image; d) "A photo of a warthog"; e) "A photo of a warthog and a traitor"; f) Heat map of the word "warthog" in generated image.

Figure 8: a) t-SNE Visualization of 100 images each of "bat", "baseball bat", "bat and ball" and text "a photo of a bat." b) The boxplot of cosine similarities between the text embedding of "a photo of a bat" and 100 of image embeddings each of "bat", "baseball bat", and "bat and ball".

The second case that we observed in our attacks is when two nouns in the text prompt share similar coarse-grained characteristics, the generated image will contain a subject that is a combination of these two nouns. As illustrated in Fig. 6, when given the text "a silver salmon and a feather," the GMs generate an image of a feather with the outline of a salmon. This happens because these two nouns (*i.e.*, salmon and feather) exhibit a certain degree of similarity in their coarse-grained attributes. In contrast, there is no feature entanglement between "salmon" and "magician" because their coarse-grained features are vastly different from each other.

To verify this assumption, we first obtain the generated latent variable for the prompt "a photo of a silver salmon" at the early sampling step (*e.g.*, 8 steps). Using this latent variable, we replace the prompt with "a photo of a feather" and continue generating images. The results confirm that feathers can continue to be generated based on the coarse-grained properties of silver salmon, and the final generated graph has high similarity with the generated graph of the prompt "a photo of a silver salmon and a feather". However, replacing "silver salmon" with "magician" does not seem to generate any object similar to "magician". This observation indicates that there is no coarse-grained feature entanglement between these two subjects. We summarize this observation in Pattern 2.

**Pattern 2** (Similarity of Coarse-grained Characteristics). *Let $X_t^A$ and $X_t^B$ denote the latent variables generated by the GM for word tokens A and B, respectively. Suppose $t$ is small and let $d$ represent the metric that measures the outline similarity between two images. If the text prompt contains both A and B, and $d(X_t^A, X_t^B)$ falls below the threshold $\sigma$, then feature entanglement occurs in the generated image.*

Based on the observed Pattern 2, The types of feature entanglement can be further divided into, direct entanglement and indirect entanglement. As shown in Fig. 5, direct entanglement represents the direct entanglement triggered by two categories of coarse-grained attributes that have global or local similarities. Indirect entanglement is shown in Fig. 5d, where the additional attribute trunk brought by the howler monkey has a high similarity with the coarse-grained attribute of the snake, thus triggering the entanglement phenomenon.

### 4.3 Polysemy of Words

**Observation 3.** *When a text prompt contains a noun A representing the object to be generated, if the semantic scope of noun A encompasses multiple distinct objects, the generated image contains one of the objects described by noun A. If there exists another word B that, when combined with noun A, directs its semantics to a particular object, the introduction of word B into the prompt, either through addition or replacement, leads to the generation of an image that specifically contains that particular object.*

The third scenario we observed in the attack is that the content of the generated image is not directly related to either the desired image or the added word. However, this is again different from a category of disappearance, where the desired target has a clear individual in the image. As illustrated in Figs. 7a, 7b and 7c, when the cleaning prompt "a photo of a bat" was modified to "a photo of a bat and a ball", the bat disappeared completely from the generated image, but the DDAM[33] heat map showed that the word "bat" is highly associated with the stick-like object in the newly generated image.

**Pattern 3** (Polysemy of Words). *When interpreting polysemous words, language models must rely on contextual cues to distinguish meanings. However, in some cases, the available contextual information may be insufficient or confuse the model by modifying specific words, resulting in a model-generated image that deviates from the actual intention of the user.*

To further investigate the phenomenon of word polysemy in the Stable diffusion model, we used the prompts "a photo of a bat", "a photo of a baseball bat" and "a photo of a bat and a ball" to generate 100 images each using the stable diffusion model, and transformed these images into embedding form by CLIP image encoder, and transformed "a photo of a bat" into embedding form by CLIP text encoder, then visualized these 301 embeddings by t-SNE. As illustrated in Fig. 8a, Considering the entire set of bat images, bats (the animal) are closer to the text "a photo of a bat" than baseball bats are, as depicted in the t-SNE visualization. However, the distribution of the two categories also has relatively close proximity, indicating their underlying similarities. The category of "bat and ball" shows a more expansive distribution, almost enveloping the other two. This suggests that by modifying the original text from "a photo of a bat" to "a photo of a bat and a ball", the distribution of the clean text can be pulled towards another meaning in the polysemous nature of the word "bat". From the perspective of text-to-image model, this kind of modification can stimulate the polysemous property of the word, thereby achieving an attack effect.

In addition to this explicit polysemy, our algorithm further demonstrates its aptitude for detecting subtler instances of polysemous words. As depicted in Figure 7, the transformative capacity of our algorithm is evident when an image of a warthog (Fig. 7d) transfigures into an image of a military chariot (Fig. 7e) with the incorporation of the term "traitor".

## 5   Auto-attack on Text-to-image Model

We aim to design an automatic attack method that targets the recent popular text-to-image models. The objective of our method is to identify attack prompts $c'$ based on a clean prompt $c$, which leads to a vision model $h : \mathcal{X} \to \mathcal{Y}$ failing to predict the desired class $y$, i.e. $\mathrm{argmax}_i\, h(\tilde{\boldsymbol{x}})_i \neq y$:

$$\boldsymbol{c}' = \underset{\boldsymbol{c}' \in B_d(\boldsymbol{c}, \xi)}{\mathrm{argmax}}\ \ell(y, h(\mathrm{GM}(\boldsymbol{c}'))) \tag{1}$$

where $\mathrm{GM}(\cdot)$ is a text-to-image model, $B(\boldsymbol{c}, \xi) = \{\boldsymbol{c}' : d(\boldsymbol{c}, \boldsymbol{c}') \leq \xi\}$, $d(\cdot, \cdot)$ is a distance measure regularizing the similarity between $\boldsymbol{c}$ and $\boldsymbol{c}'$, and $\xi$ is a maximum distance. To enable auto-attack, a differentiable method that can be optimized using gradient descent is desired. We introduce a Gumbel-Softmax sampler to enable differentiable modifications on text prompts during the word embedding phase. To minimize the distance $d(\boldsymbol{c}, \boldsymbol{c}')$, we introduce two constraints, including a fluency constraint and a similarity constraint.

In our experimental setup, the open-source Stable Diffusion model is employed as the targeted generative model $\mathrm{GM}(\cdot)$. By generating white-box attack prompts for Stable Diffusion, we can subsequently transfer these prompts to other generative models to execute black-box attacks. To facilitate the classification task, we utilize a CLIP classifier as the vision model $h(\cdot)$, benefiting from its exceptional zero-shot classification accuracy. To establish the desired classes, we employ the 1,000 classes derived from ImageNet-1K. In the case of generating short prompts, a fixed template of "A photo of [CLASS_NAME]" is utilized to generate the prompts. Conversely, for the generation of long prompts, we employ ChatGPT 4 [24] as a prompt generation model. Subsequently, human experts verify the correctness of the prompts and check that the prompts indeed contain the noun associated with the desired class.

### 5.1   Differentiable Text Prompt Modification

A text prompt is typically a sequence of words $\boldsymbol{c} = \{\boldsymbol{c}_1, \ldots, \boldsymbol{c}_K\}$. Owing to the discrete nature of text prompts $\boldsymbol{c}$, perturbations can be incorporated either by replacing an existing word $\boldsymbol{c}_k$ where $1 \leq k \leq K$ or augmenting with new ones $\{\boldsymbol{c}_{K+i} | 1 \leq i \leq K'\}$. However, the non-differentiable nature of this procedure makes it unsuitable for optimization utilizing gradient-based techniques. Therefore, there is a need for a mechanism that guarantees the differentiability of the word selection process. In this regard, we integrate a Gumbel Softmax sampler $\psi(\cdot; \tau)$ into the word embedding phase. The Gumbel Softmax function has the ability to approximate a one-hot distribution as the temperature $\tau \to 0$. Additionally, the Gumbel distribution has the ability to introduce further randomness, thereby enhancing the exploitability during the initial stages of perturbation search.

**Differentiable Sampling.** We employ a trainable matrix $\boldsymbol{\omega} \in \mathbb{R}^{K \times V}$ to learn the word selection distribution, where $K$ is the length of the text prompt and $V$ is the vocabulary size. In the scenario of augmenting with a new word, the sequence length $K$ can be extended to $K + K'$ to facilitate the addition of new words. The Gumbel Softmax can be represented as follows:

$$\text{GumbelSoftmax}(\boldsymbol{\omega}_k; \tau) := \frac{\exp((\boldsymbol{\omega}_{k,i} + g_{k,i})/\tau)}{\sum_j \exp((\boldsymbol{\omega}_{k,j} + g_{k,j})/\tau)}, \tag{2}$$

where $g_{k,i} \sim \text{Gumbel}(0, 1)$ are i.i.d. samples from a Gumbel distribution. The word embedding stage employs a matrix $E \in \mathbb{R}^{V \times D}$, where $V$ is the vocabulary size and $D$ is for the embedding dimensionality. The discrete process of word embedding is to select $E_i$ from $E$ based on the index $1 \leq i \leq V$ of a word in the vocabulary. To make this process differentiable , the dot product between $\text{GumbelSoftmax}(\boldsymbol{\omega}_k)$ and $E$ can be calculate:

$$\boldsymbol{c}'_k = \psi(\boldsymbol{\omega}_k; \tau) = \text{GumbelSoftmax}(\boldsymbol{\omega}_k; \tau) \cdot E \approx E_i, \quad \text{s.t. } i = \underset{i}{\arg\max} \, \boldsymbol{\omega}_{k,i}, \tag{3}$$

where $\boldsymbol{c}'_k$ is the new word selected according to $\boldsymbol{\omega}_k$. As $\tau \to 0$, Eq. 3 can effectively emulate an $\arg\max$ selection procedure.

Additionally, in order to ensure similarity, it is desired to preserve the noun representing the desired object in the new prompt. This is achieved by utilizing a binary mask $M \in \{0, 1\}^K$, where the position corresponding to the desired noun is set to $0$ while other positions are set to $1$. By computing $\boldsymbol{c}' \leftarrow (1 - M) \cdot \boldsymbol{c} + M \cdot \boldsymbol{c}'$, the desired noun can be retained in the prompt while other words can be modified.

**Attack Objective.** To generate images $\tilde{\boldsymbol{x}}$ that cannot be correctly classified by the classifier, a margin loss [1] can be used as the loss function $\ell(\cdot, \cdot)$ in Eq. 1:

$$\ell_{\text{margin}}(\tilde{\boldsymbol{x}}, y; h) = \max\left(0, h(\tilde{\boldsymbol{x}})_y - \max_{i \neq y} h(\tilde{\boldsymbol{x}})_i + \kappa\right), \tag{4}$$

where $\kappa$ is a margin. Eq. 4 reduces the classifier's confidence on the true class $y$ and improve its confidence on the class with the largest confidence, excluding $y$ until a margin $\kappa$ is reached.

### 5.2 Constraints on Fluency and Similarity

Given that we search for perturbations in a $\mathbb{R}^{K \times V}$ space to attack the text prompt, the attack prompts may be too diverse if the added perturbations are not properly constrained, making it easily detectable. Eq. 1 includes a distance constraint such that $d(\boldsymbol{c}, \boldsymbol{c}') \leq \xi$, which ensures that the added perturbations are subtle and hard to notice. The measurement of distance between two pieces of text can be approached through various methods. We introduce two constraints to reduce this distance, namely a **fluency** constraint and a **semantic similarity** constraint. The fluency constraint ensures that the generated sentence is smooth and readable, while the semantic similarity semantic constraint regularize the semantic changes introduced by the perturbations, making the attack prompt $\boldsymbol{c}'$ closely resemble the clean prompt $\boldsymbol{c}$.

**Fluency constraint.** The fluency constraint can be achieved visa a Casual Language Model (CLM) $\phi$ with log-probability outputs [9]. The next token distribution we learn is compared with the next token distribution predicted by $\phi$. Given a sequence of perturbed text $\boldsymbol{c}'$, we use $\phi$ to predict the a token $\boldsymbol{c}'_i$ based on $\{\boldsymbol{c}'_1, \ldots \boldsymbol{c}'_{i-1}\}$. Therefore, we can have a log-likelihood of the possible next word:

$$\log p_\phi\left(\boldsymbol{c}'_i | \boldsymbol{c}'_1, \ldots, \boldsymbol{c}'_{i-1}\right) = \phi(\boldsymbol{c}'_1, \ldots \boldsymbol{c}'_{i-1}). \tag{5}$$

The next token distribution we learn can be easily obtained by $\text{GumbelSoftmax}(\boldsymbol{\omega}_i; \tau)$. Subsequently, a cross-entropy loss function can be employed to optimize the learned distribution:

$$\text{CE}_\phi(\boldsymbol{\omega}) = -\sum_{i=1}^{K} \sum_{j=1}^{D} \text{GumbelSoftmax}(\boldsymbol{\omega}_i; \tau)_j \cdot \phi(\psi(\boldsymbol{\omega}_1; \tau), \ldots \psi(\boldsymbol{\omega}_{i-1}; \tau))_j. \tag{6}$$

Eq. 6 serves as a regularizer to encourage the next word selection distribution to resemble the prediction of the CLM $\phi$, thereby ensuring fluency.

**Algorithm 1** Auto-attack on Text-to-image Models (ATM)

---

**Input:** The maximum number of iterations $T$. The maximum number of attack candidates $N$. The clean prompt $c$. The desired class $y$. A binary mask $M$. A learning rate $\eta$.
**Output:** A set of attack prompts $\mathcal{S}$.

 1: Initialize $\boldsymbol{\omega}$
 2: **for** $t = 1 \rightarrow T$ **do**                ▷ The search stage
 3:   Sample an attack prompt $\boldsymbol{c}' = \{\psi(\boldsymbol{\omega}_k; \tau) | 1 \leq k \leq K\}$
 4:   Apply the mask by $\boldsymbol{c}' \leftarrow (1 - M) \cdot \boldsymbol{c} + M \cdot \boldsymbol{c}'$
 5:   Generate an image $\tilde{\boldsymbol{x}}' = \mathrm{GM}(\boldsymbol{z}_T | \boldsymbol{c}')$
 6:   Get classification results $y' = h(\tilde{\boldsymbol{x}}')$
 7:   Conduct a gradient descent step $\boldsymbol{\omega} \leftarrow \boldsymbol{\omega} - \eta \cdot \nabla_{\boldsymbol{\omega}} \mathcal{L}(\boldsymbol{\omega})$
 8: **end for**
 9: Initialize $\mathcal{S} = \varnothing$
10: **for** $n = 1 \rightarrow N$ **do**                ▷ The attack stage
11:   Sample an attack prompt $\boldsymbol{c}' = \{\psi(\boldsymbol{\omega}_k; \tau) | 1 \leq k \leq K\}$
12:   Apply the mask by $\boldsymbol{c}' \leftarrow (1 - M) \cdot \boldsymbol{c} + M \cdot \boldsymbol{c}'$
13:   Generate an image $\tilde{\boldsymbol{x}}' = \mathrm{GM}(\boldsymbol{z}_T | \boldsymbol{c}')$
14:   **if** $\arg\max h(\tilde{\boldsymbol{x}}') \neq y$ **then**          ▷ If attack success
15:    Save the success attack prompt $\mathcal{S} \leftarrow \mathcal{S} \cup \{\boldsymbol{c}'\}$
16:   **end if**
17: **end for**

---

**Semantic similarity constraint.** Rather than simply considering a word similarity, we concern more about semantic similarity. One prominent metric used to evaluate semantic similarity is the BERTScore [39]. The calculation of BERTScore requires contextualized word embeddings. The aforementioned CLM $\phi$ is used again to extract the embeddings $\boldsymbol{v} = \phi^{(\mathrm{emb})}(\boldsymbol{c})$, where $\phi^{(\mathrm{emb})}$ denotes the embedding network used in $\phi$. The BERTScore between the clean prompt $\boldsymbol{c}$ and the attack prompt $\boldsymbol{c}'$ can be calcualted by

$$S_{\mathrm{BERT}}(\boldsymbol{c}, \boldsymbol{c}') = \sum_{i=1}^{N} w_i \max_{j=1,\dots,M} \boldsymbol{v}_i^\top \boldsymbol{v}_j', \tag{7}$$

where $w_i := \mathrm{idf}(\boldsymbol{c}_i) / \sum_{i=1}^{N} \mathrm{idf}(\boldsymbol{c}_i)$ is the normalized inverse document frequency (idf), $N = K$, and $M$ is either $K$ or $K + K'$ depending on whether existing words are being replaced or new words are being added. To improve the similarity, we use $1 - S_{\mathrm{BERT}}(\boldsymbol{c}, \boldsymbol{c}')$ as the our loss term.

**The constrained objective.** Considering that the addition of constraints may limit the diversity of perturbation search, we introduce two hyper-parameters, $\lambda$ and $\gamma$, to control the strength of the constraints. Then, the overall objective function can be written as:

$$\max_{\boldsymbol{\omega}} \quad \mathbb{E}_{\boldsymbol{z}_T \sim \mathcal{N}(0,1), \boldsymbol{c}' = \psi(\boldsymbol{\omega}; \tau)} \left[ \ell_{\mathrm{margin}}(\mathrm{GM}(\boldsymbol{z}_T | \boldsymbol{c}'), y; h) \right] \tag{8}$$

$$\textbf{s.t.} \quad \min_{\boldsymbol{\omega}} \quad \lambda \cdot \mathrm{CE}_\phi(\boldsymbol{\omega}) + \gamma \cdot (1 - S_{\mathrm{BERT}}(\boldsymbol{c}, \boldsymbol{c}')). \tag{9}$$

### 5.3 Generation of Attack Prompts

The overall procedure of ATM is as described in Algorithm 1. It consists of two stages: a search stage, where the Gumbel Softmax distribution is learned, and an attack stage, where we generate attack prompts using the learned distribution. In the search stage, we use gradient descent to optimize the parameters $\boldsymbol{\omega}$ for each clean prompt $\boldsymbol{c}$ over a period of $T$ iterations. Once $\boldsymbol{\omega}$ is learned, we proceed to the attack stage. In this stage, we sample $N$ attack prompts from each learned $\boldsymbol{\omega}$. An attack prompt $\boldsymbol{c}'$ is considered successful if the image $\tilde{\boldsymbol{x}}'$ generated from it cannot be correctly classified by the visual classifier $h$.

### 5.4 Attack Experiment Results

To corroborate the effectiveness of our proposed algorithm, we implemented a structured evaluation scheme based on the model template: "A photo of [CLASS_NAME]". An extensive dataset comprising one thousand abbreviated textual descriptions, corresponding to all classes within ImageNet [3], was meticulously curated. Additionally, comprehensive long-form textual descriptions for these

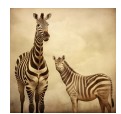 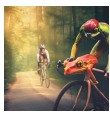 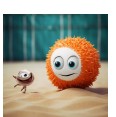 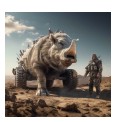 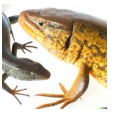 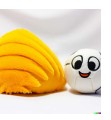 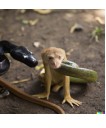 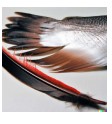

| (a) | (b) | (c) | (d) | (e) | (f) | (g) | (h) |

Figure 9: (a),(b),(c),(d) are generated by Midjourney, and the corresponding prompts are, "a photo of a zebra and a giraffe"; "a photo of a tree frog and a cyclist"; "a photo of a volleyball and a Nemo"; and "a photo of a warthog and a traitor", respectively. (e),(f),(g),(h) are generated by DALL·E2, and the corresponding prompts are "a photo of a tench and a lizard"; "a photo of a volleyball and a Nemo"; "A photo of a howler monkey and a snake"; and "a photo of a silver salmon and a feather", respectively.

same classes were generated utilizing the capabilities of ChatGPT4 [24]. Upon rigorous testing, our algorithm demonstrated noteworthy performance, recording attack success rates of 91.1% and 81.2% on the short-text and long-text datasets, respectively. Moreover, the calculated cosine similarity between the embeddings of the original and attack prompts yielded values of 0.7227 and 0.8364, respectively. Due to the length of this paper, detailed experiment results, including comparisons to baselines and ablation studies, are reported in the supplementary material.

To further investigate whether our generated attack prompts can be transferred to different text-to-image models, we randomly select several attack prompts to attack DALL·E2 and Midjourney, respectively. The experimental results prove that our attack prompts can also be used for black-box attacks. More results of black-box attacks are reported in the supplementary material.

## 6 Conclusion

The realm of text-to-image generation has observed a remarkable evolution over recent years, while concurrently exposing several vulnerabilities that require further exploration. Despite the many advancements, there are key limitations, specifically concerning the stability and reliability of generative models, which remain to be addressed. This paper has introduced Auto-attack on Text-to-image Models (ATM), a novel approach that generates a plethora of successful attack prompts, providing an efficient tool for probing vulnerabilities in text-to-image models. ATM not only identifies a broader array of attack patterns but also facilitates a comprehensive examination of the root causes. We believe that our proposed method will inspire the research community to shift their focus toward the vulnerabilities of present-day text-to-image models, stimulating further exploration of both attack and defensive strategies. This process will be pivotal in advancing the security mechanisms within the industry and contributing to the development of more robust and reliable systems for generating images from textual descriptions.

## 7 Acknowledgments

This work was supported in part by the Australian Research Council under Projects DP210101859 and FT230100549.

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

## Supplementary Material

In this supplementary material, we first delve into additional analyses concerning the vulnerabilities observed in the Stable Diffusion model (Section A). Subsequently, we offer instances of long and short prompt attacks, accompanied by the corresponding generated images, as well as instances of black-box attacks (Section B). Lastly, we undertake a comprehensive series of experiments to substantiate the effectiveness of our approach (Section C). These experiments include the evaluation of attacks targeting both long and short prompts. Additionally, ablation studies are conducted to explore attacks employing different search steps, assess the influence of our fluency and semantic similarity constraints on text similarity, and target diverse samplers (e.g., DDIM and DPM-Solver) in the attack process.

## A Vulnerabilities of Stable Diffusion Model

### A.1 Pattern 1: Variability in Generation Speed

Fig. 10 demonstrates the entire 50-step violin diagram which has been discussed before. To eliminate possible bias due to a single metric, we further verified the difference in generation speed of one thousand images based on the LPIPS [38] metric, as shown in Fig. 11, The calculation of the LPIPS distance from the images generated at each stage to the ultimate image is performed. The horizontal axis signifies the range of steps from 49 down to 0, whereas the vertical axis denotes the respective LPIPS scores. Each violin plot illustrates the distribution of the LPIPS scores associated with 1,000 images at a specific step. The width of the violin plot is proportional to the frequency at which images achieve a certain score. During the initial stages of generation, the distribution's median is situated nearer to the maximum LPIPS value, suggesting a preponderance of classes demonstrates slower generation velocities. Nonetheless, the existence of a low minimum value indicates the presence of classes that generate at comparatively faster rates. As the generation transitions to the intermediate stages, the distribution's median progressively decreases, positioning itself between the maximum and minimum LPIPS values. In the concluding stages of generation, the distribution's median is found closer to the minimum LPIPS value, implying that the majority of classes are nearing completion. However, the sustained high maximum value suggests that there are classes still exhibiting slower generation rates.

### A.2 Pattern 2: Similarity of Coarse-grained Characteristics

To further verify that coarse-grained feature similarity is the root cause of feature entanglement, we provide more cases in Fig.12. From these cases, it is evident that for any two classes prone to feature entanglement, they can each proceed with image generation based on the other's coarse-grained information.

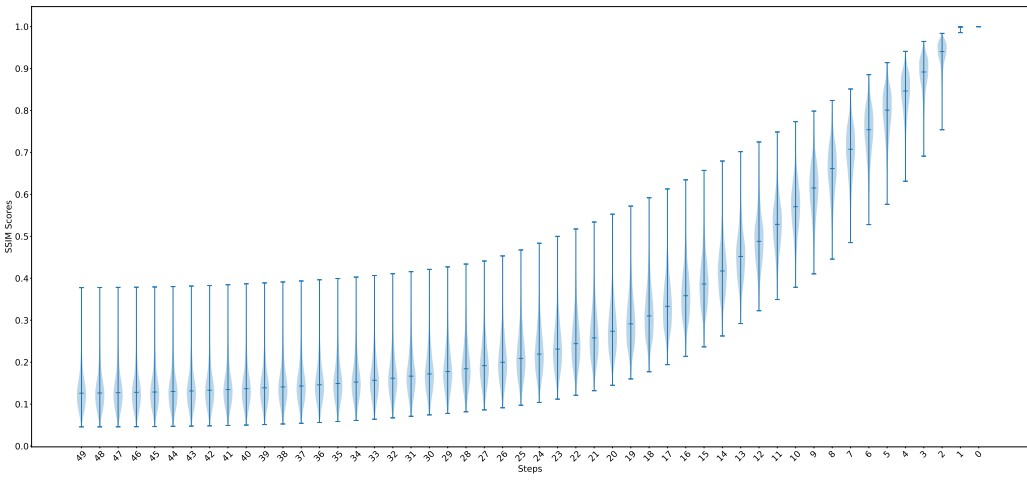

Figure 10: A violin plot illustrating the generation speeds of $1,000$ images of various classes. The horizontal axis represents the number of steps taken, ranging from 49 to 0, while the vertical axis displays the SSIM scores. The width of each violin represents the number of samples that attained a specific range of SSIM scores at a given step.

## A.3 Pattern 3: Polysemy of Words

As shown in Fig. 13, when we attack the prompt "A photo of a warthog" to "A photo of a warthog and a traitor", the original animal warthog becomes an object similar to a military vehicle or military aircraft, while the images generated by attack prompt is not directly related to the image of the animal warthog or traitor. From the t-SNE visualization (Fig. 14), we can see that the distance from the picture generated by the attack prompt to the text "a photo of a warthog" has a similar distance to the animal warthog picture to the text, so we can see that by attacking the original category word that guided the original category word (animal warthog) into its alternative meaning.

From the box plots (Fig. 15), it can be observed that the image of "warthog" exhibits the highest similarity with the prompt's embedding, while the image of "traitor" demonstrates the lowest similarity, as anticipated. Simultaneously, the similarity distribution between the images of "warthog" and "traitor" with the prompt text is relatively wide, indicating that some images have a high similarity with "warthog," while others lack features associated with "warthog."

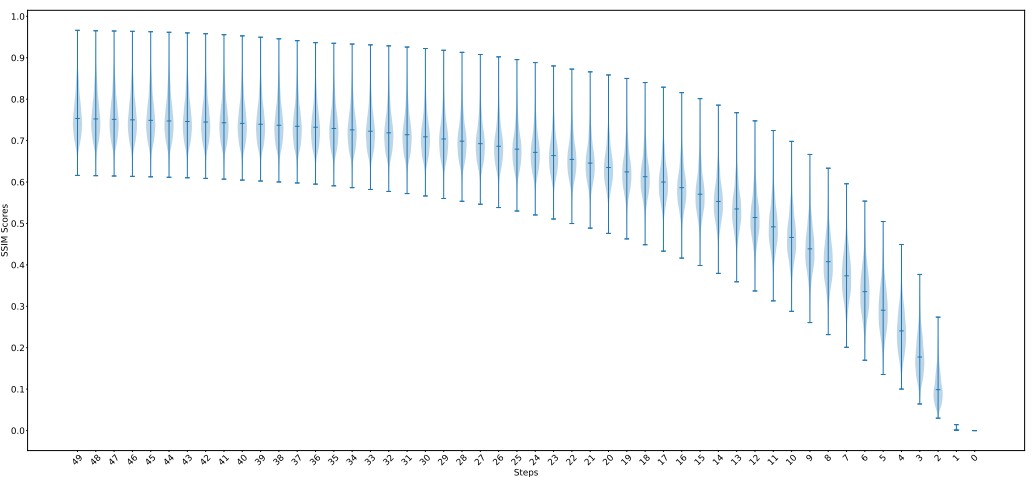

Figure 11: A violin plot illustrating the generation speeds of $1,000$ images of various classes. The horizontal axis represents the number of steps taken, ranging from 49 to 0, while the vertical axis displays the LPIPS scores. The width of each violin represents the number of samples that attained a specific range of LPIPS scores at a given step.

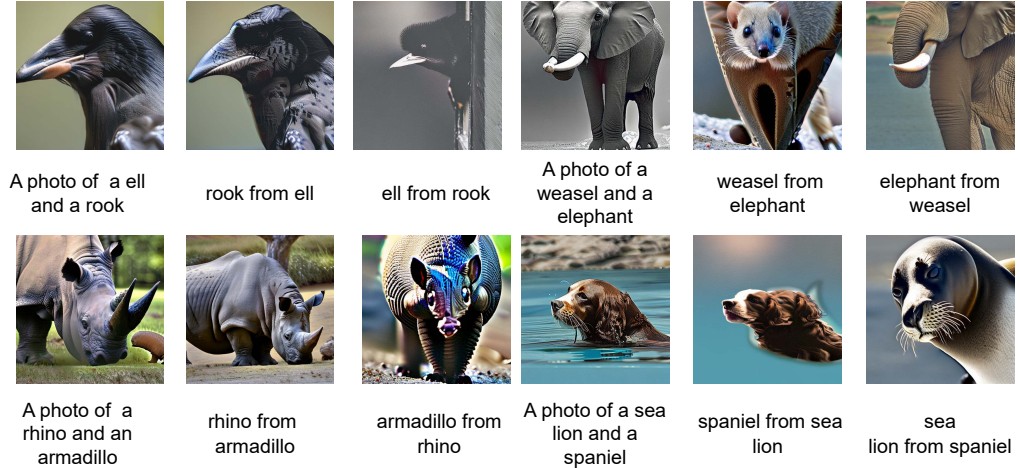

Figure 12: The image caption, "A photo of class$A$ and class$B$" represents the generated image when feature entanglement occurs; And "class$A$ from class$B$" represents the final generated image of prompt "A photo of class$A$" based on the eighth step of the prompt "A photo of class$B$"

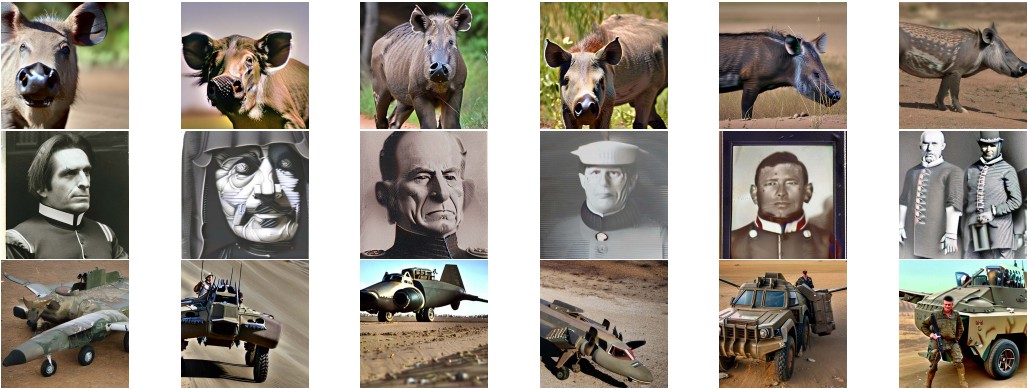

Figure 13: The images in the first row are generated by the prompt "A photo of a warthog". The images in the second row are generated by the prompt "A photo of a traitor". The images in the third row are generated by the prompt "A photo of a warthog and a traitor".

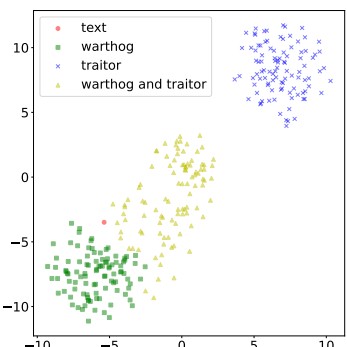

Figure 14: t-SNE Visualization of 100 images each of "warthog", "traitor", "warthog and traitor" and text "a photo of a warthog."

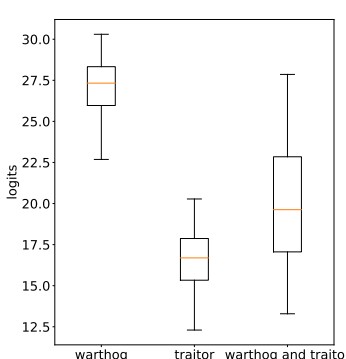

Figure 15: The boxplot of cosine similarities between the text embedding of "a photo of a warthog" and 100 of image embeddings each of "warthog", "traitor", and "warthog" and traitor".

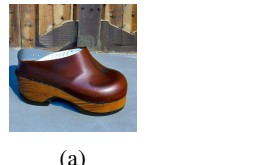          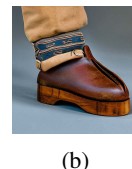          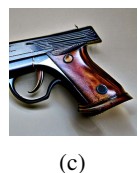

(a)                    (b)                    (c)

Figure 16: a) "A type of footwear with a thick, rigid sole, often made of wood, and an upper made of leather or another material. Clogs can be open or closed, and are commonly associated with Dutch and Scandinavian cultures." b) "footwear" is replaced by "pistol". c) "Dutch" is replaced by "pistol".

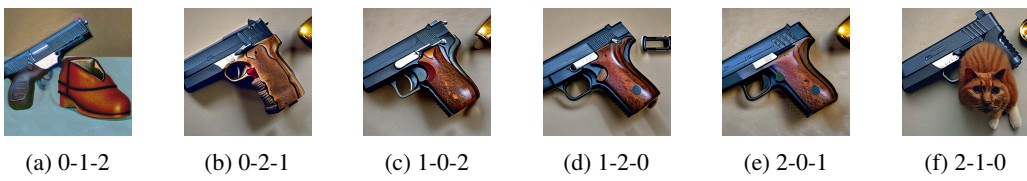

(a) 0-1-2     (b) 0-2-1     (c) 1-0-2     (d) 1-2-0     (e) 2-0-1     (f) 2-1-0

Figure 17: A template, "A photo of $A$, $B$ and $C$", is used to generate prompts, where $A, B, C \in$ {"cat", "pistol", "clogs"}. For exmaple, "0-1-2" represents $A$ = "cat", $B$ = "pistol" and $C$ = "clogs", and so on.

### A.4 Pattern 4: Positioning of Words

In addition to the three aforementioned observations and patterns outlined in the paper, there is a fourth observations (Observation 4), which is related to positioning of words.

**Observation 4.** *When a text prompt contains a noun $A$ representing the object to be generated, there exists a preceding word $B$ and a succeeding word $C$ around noun $A$. When replacing either word $B$ or $C$ with another noun $D$, for certain instances of noun $A$, replacing word $B$ results in the generation of an image containing noun $D$, while replacing word $C$ still results in the generation of an image containing noun $A$. Conversely, for other instances of noun $A$, the opposite scenario occurs.*

An example of Pattern 4 is shown in Fig. 16. When "footwear" is replaced by "pistol", the generated image contains a pistol instead of clogs. However, when "Ductch" is replaced by "pistol", the model still generates an image of clogs. In addition to differences in the words being replaced, a significant distinction between the two aforementioned examples of success and failure lies in the relative positioning of the word being replaced with respect to the target class word. We hypothesize that this phenomenon occurs due to the different order of the replaced words $B$ or $C$ with respect to the noun $A$. To exclude the effects of complex contextual structures, a template for a short prompt, "A photo of $A$, $B$ and $C$", is used, and the order of $A$, $B$, and $C$ are swapped (Fig. 17).

When these sentences with different sequences of category words are understood from a human perspective, they all have basically the same semantics: both describe a picture containing a cat, clogs, and a pistol. However, in the processing of language models (including CLIP), the order of words may affect their comprehension. Although positional encoding provides the model with the relative positions of words, the model may associate different orders with different semantics through learned patterns. Therefore, we propose our Pattern 4.

**Pattern 4** (Positioning of Words). *Let $\mathcal{V}$ denote a set of vocabulary. Let $\mathcal{N} \subset \mathcal{V}$ denote the subset of all nouns in the vocabulary. Consider a text prompt containing noun $A \in \mathcal{N}$ representing the object to be generated. Furthermore, assume there exist preceding word $B \in \mathcal{V}$ and succeeding word $C \in \mathcal{V}$ surrounding noun $A$. There exists a condition-dependent behavior regarding the replacement of words $B$ and $C$ with another noun $D \in \mathcal{N}$:*

$$
\begin{cases}
\exists A, D \in \mathcal{N}, \quad \exists B, C \in \mathcal{V}, \quad P(B \to D) \xRightarrow{\text{generate}} D \ \bigwedge \ P(C \to D) \xRightarrow{\text{generate}} A; \\
\exists A, D \in \mathcal{N}, \quad \exists B, C \in \mathcal{V}, \quad P(B \to D) \xRightarrow{\text{generate}} A \ \bigwedge \ P(C \to D) \xRightarrow{\text{generate}} D.
\end{cases}
$$

## B Cases of Short/Long-Prompt Attacks and Black-box Attacks

### B.1 Attack on Long Prompt

In Fig. 18, we demonstrate more cases of long text prompt attacks.

### B.2 Attack on Short Prompt

In Fig. 19, we demonstrate more cases of long text prompt attacks.

### B.3 Black-box Attack

In Fig. 20, and Fig. 21, we demonstrate black box attacks targeting Midjourney and DALL·E2, respectively.

## C Experiments

In our experiments, we conduct comprehensive analyses of both long and short prompts. Furthermore, we conduct ablation studies specifically on long prompts, focusing on three key aspects. Firstly, we evaluate our attack method with different numbers of search steps $T$. Secondly, we investigate the influence of our constraints, including fluency and semantic similarity as measured by BERTScore. Lastly, we attack different samplers, including DDIM [31] and DPM-Solver [17].

### C.1 Experimental Setting.

**Attack hyperparamters.** The number of search iterations $T$ is set to 100. This value determines the number of iterations in the search stage, during which we aim to find the most effective attack prompts. The number of attack candidates $N$ is set to 100. This parameter specifies the number of

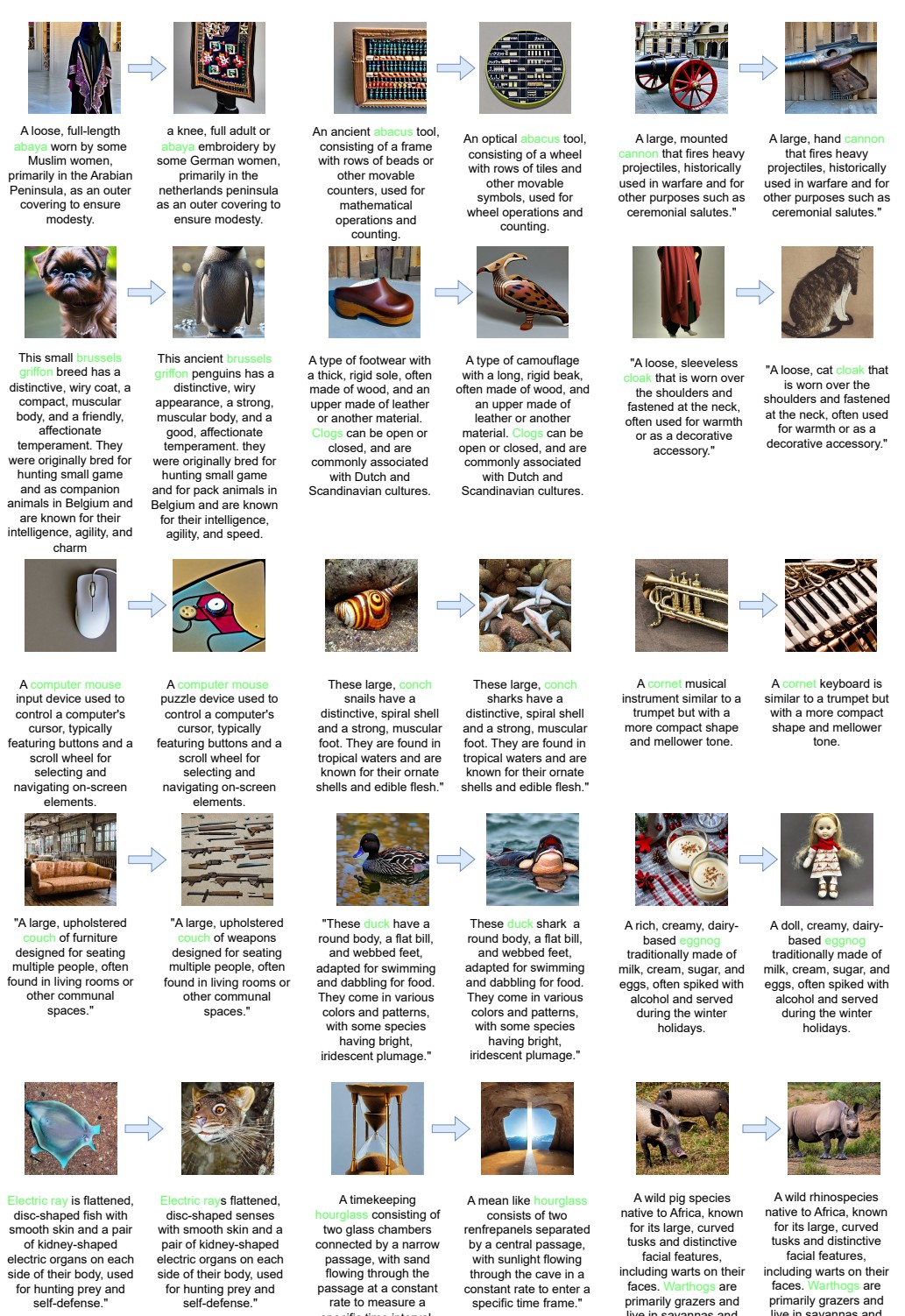

Figure 18: To the left of the arrow is the clean long text prompt (highlighted by green) and its corresponding image, to the right of the arrow is the generated attack prompt (highlighted by red) and its corresponding image. (Section B.1 Attack on Long Text Prompt)

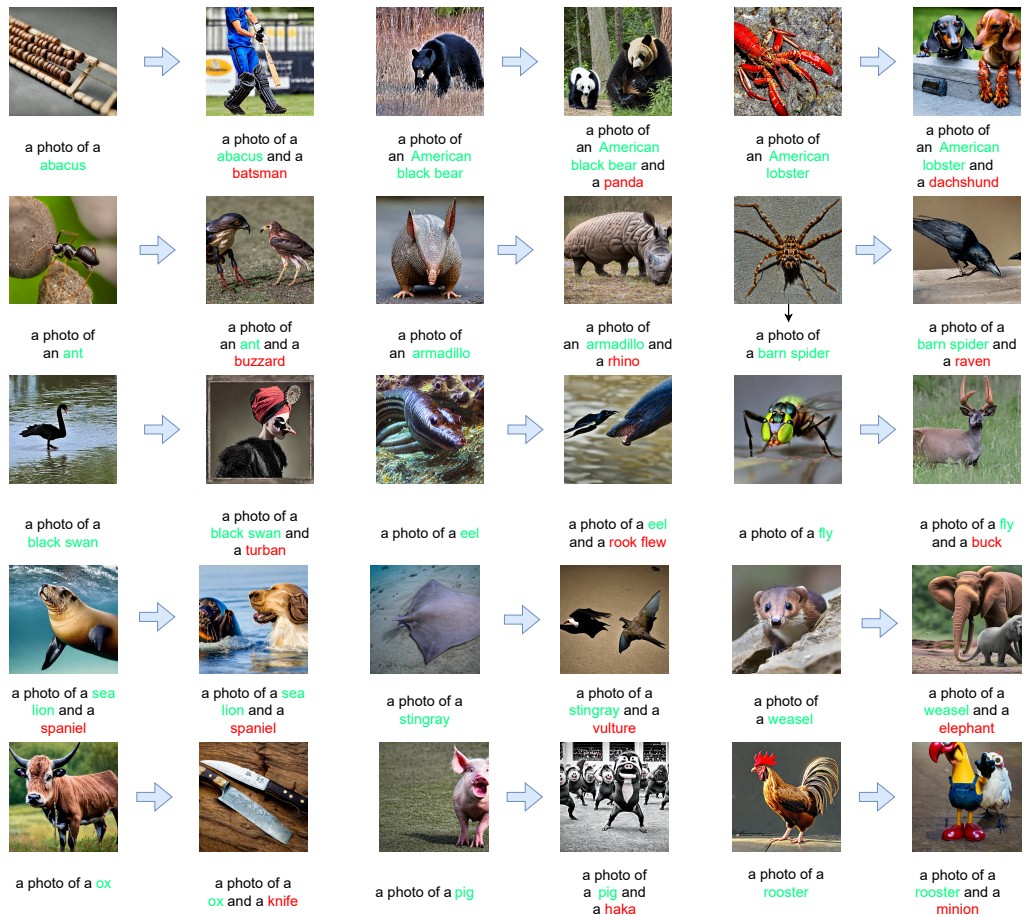

Figure 19: To the left of the arrow is the clean short text prompt (highlighted by green) and its corresponding image, to the right of the arrow is the generated attack prompt (highlighted by red) and its corresponding image. (Section B.2 Attack on Short Text Prompt)

candidate attack prompts considered in the attack stage, allowing for a diverse range of potential attack prompts to be explored. The learning rate $\eta$ for the matrix $\boldsymbol{\omega}$ is set to $0.3$. The margin $\kappa$ in the margin loss is set to $30$.

**Text prompts.** Our experiments consider the $1,000$ classes from ImageNet-1K [3], which serves as the basis for generating images. To explore the impact of prompt length, we consider both short and long prompts. For clean short prompts, we employ a standardized template: "A photo of [CLASS_NAME]". Clean long prompts, on the other hand, are generated using ChatGPT 4 [24], with a prompt length restriction of 77 tokens to align with the upper limit of the CLIP [25] word embedder.

**Evaluation metrics.** To evaluate the effectiveness of our attack method, we generate attack prompts from the clean prompts. We focus on three key metrics: **success rate**, Fréchet inception distance [11] (**FID**), Inception Score (**IS**), and text similarity (**TS**). Subsequently, $50,000$ images are generated using the attack prompts, ensuring a representative sample of 50 images per class. The success rate is determined by dividing the number of successful attacks by the total of 1,000 classes. FID and IS are computed by comparing the generated images to the ImageNet-1K validation set with (**torch-fidelity**)[22]. TS is calculated by embedding the attack prompts and clean prompts using the CLIP [25] word embedder, respectively. Subsequently, the cosine similarity between the embeddings is computed to quantify the text similarity.

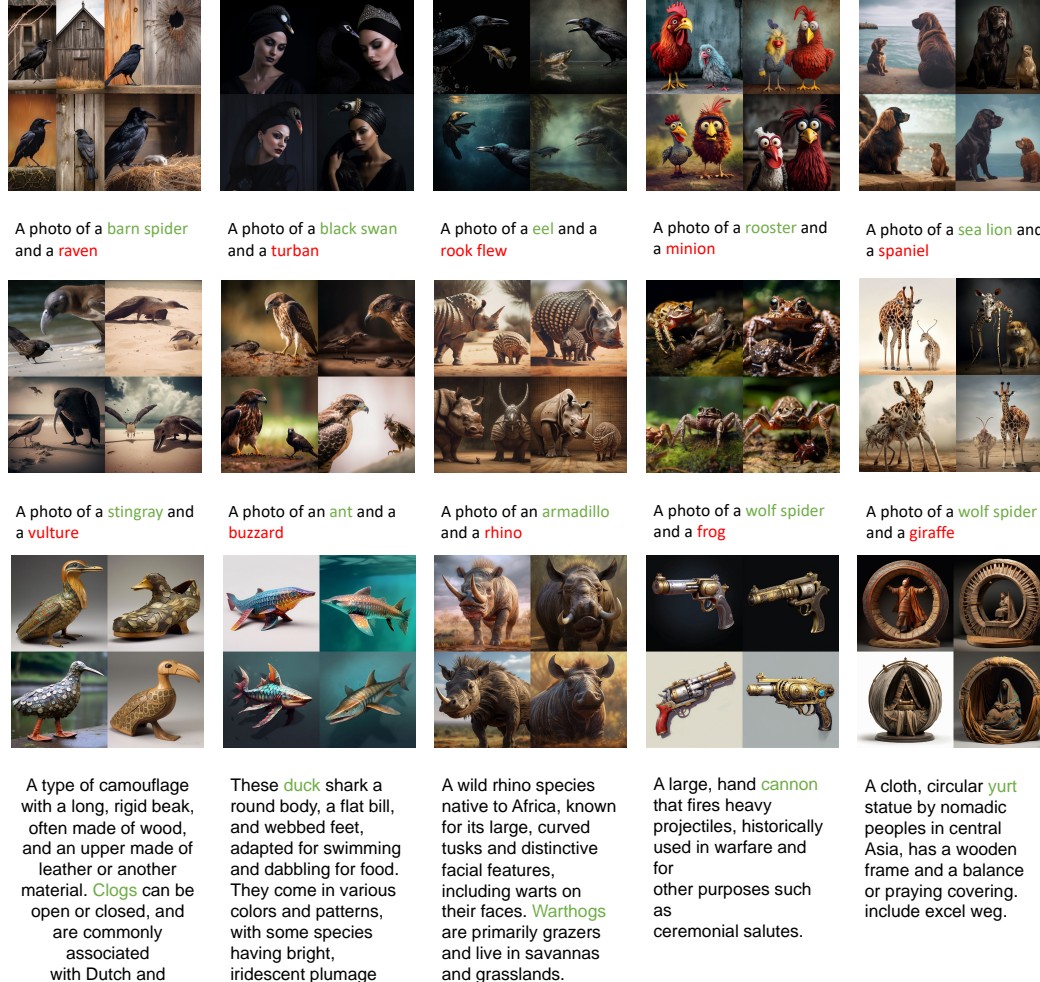

Figure 20: Black-box attack on Midjourney (Section B.3 Black-box Attack).

Table 1: Main results of short-prompt and long-prompt attacks.

| Prompt | Method | Success (%) | FID ($\downarrow$) | IS ($\uparrow$) | TS ($\uparrow$) |
|--------|--------|-------------|--------|-----|-----|
| **Short** | Clean | - | 18.51 | 101.33±1.80 | 1.00 |
| | Random | 79.2 | 29.21 | 66.71±0.87 | 0.69 |
| | ATM (Ours) | 91.1 | 30.09 | 65.98±1.10 | 0.72 |
| **Long** | Clean | - | 17.95 | 103.59±1.68 | 1.00 |
| | Random | 41.4 | 24.16 | 91.33±1.58 | 0.94 |
| | ATM (Ours) | 81.2 | 29.65 | 66.09±1.83 | 0.84 |

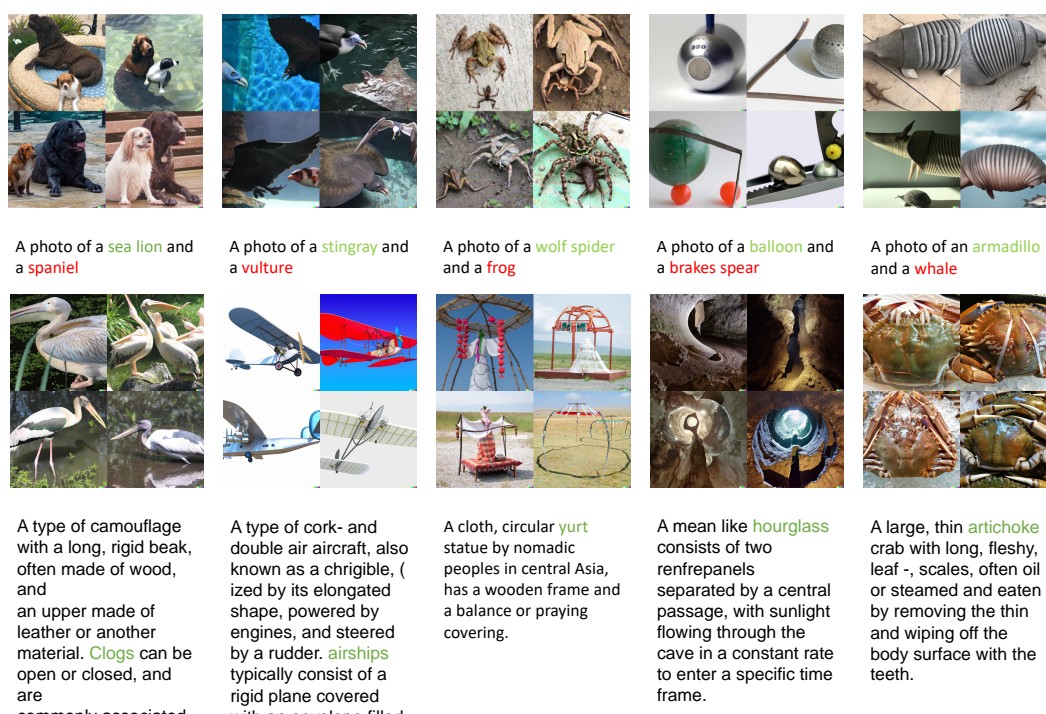

A photo of a sea lion and a spaniel

A photo of a stingray and a vulture

A photo of a wolf spider and a frog

A photo of a balloon and a brakes spear

A photo of an armadillo and a whale

A type of camouflage with a long, rigid beak, often made of wood, and an upper made of leather or another material. Clogs can be open or closed, and are commonly associated with Dutch and Scandinavian cultures.

A type of cork- and double air aircraft, also known as a chrigible, ( ized by its elongated shape, powered by engines, and steered by a rudder. airships typically consist of a rigid plane covered with an envelope filled with oxygen, such as helium or hydrogen.

A cloth, circular yurt statue by nomadic peoples in central Asia, has a wooden frame and a balance or praying covering.

A mean like hourglass consists of two renfrepanels separated by a central passage, with sunlight flowing through the cave in a constant rate to enter a specific time frame.

A large, thin artichoke crab with long, fleshy, leaf -, scales, often oil or steamed and eaten by removing the thin and wiping off the body surface with the teeth.

Figure 21: Black-box attack on DALL·E2 (Section B.3 Black-box Attack).

## C.2    Main Results

Table 1 reports our main results, including short-prompt and long-prompt attacks. Compares to long text prompts, short text prompts comprise only a small number of tokens. This leads to a relatively fragile structure that is extremely vulnerable to slight disturbance. Therefore, random attacks can reach an impressive success rate of 79.2% targeting short prompts but a low success rate of 41.4% targeting the long prompts. In the contrast, our algorithm demonstrates its true potential, reaching an impressive success rate of 91.1% and 81.2% targeting short and long prompts, respectively.

As a further evidence of the effectiveness of our algorithm, it's worth noting the text similarity (TS) metrics between the random attacks and our algorithm's outputs. For short-prompt attack, the values stand at 0.69 and 0.72, respectively, illustrating that the semantic information of short texts, while easy to disrupt, can be manipulated by a well-designed algorithm with fluency and semantic similarity constraints. Our attacks preserve more similarity with the clean prompts. For long-prompt attacks, the TS score of random attacks (0.94) is higher compared to our attacks (0.84). One possible reason is that random attacks tend to make only minimal modifications as the length of the prompt increases. This limited modification can explain the significantly lower success rate of random attacks on longer prompts.

From the perspective of evaluating the quantitative metrics of generated images. For short and long texts, images generated from the clean text have the lowest FID (18.51 and 17.95) and the highest IS ($101.33\pm1.80$ and $103.59\pm1.68$). As the attack success rate rises, FID shows an upward trend. Examining this situation from the perspective of FID, a metric that gauges the distance between the distribution of generated images and the original data set. As the attack becomes more successful, the image set generated by the attack prompt tends to deviate substantially from the distribution of the original data set. This divergence consequently escalates the FID score, indicating a larger distance between the original and generated distributions. On the other hand, the attack objective of our algorithm, namely the margin loss, is used to reduce the classifier's confidence in the true class $y$ and improve its confidence in the class with the largest confidence excluding $y$, until a margin $\kappa$ is reached. Consequently, generating low-quality images is also an objective of our attack. These images contribute to the increase in the FID index and the decrease in the IS score.

Table 2: Results of the ablation study on the number of steps in attack prompt search.

| #Steps | Success (%) | FID ($\downarrow$) | IS ($\uparrow$) | TS ($\uparrow$) |
|---|---|---|---|---|
| 50 | 68.7 | 34.00 | 93.94±1.84 | 0.97 |
| 100 | 81.2 | 29.65 | 66.09±1.83 | 0.84 |
| 150 | 67.2 | 45.23 | 58.51±0.79 | 0.82 |

Table 3: Results of the ablation study on the constraints

| Fluency | BERTScore | Success (%) | FID ($\downarrow$) | IS ($\uparrow$) | TS ($\uparrow$) |
|---|---|---|---|---|---|
| ✗ | ✗ | 91.3 | 39.14 | 47.21±1.25 | 0.37 |
| ✓ | ✗ | 81.7 | 29.37 | 64.93±1.57 | 0.79 |
| ✗ | ✓ | 89.8 | 39.93 | 46.94±0.99 | 0.51 |
| ✓ | ✓ | 81.2 | 29.65 | 66.09±1.83 | 0.84 |

## C.3 Different Search Steps

Table 2 presents the results of using different numbers of steps $T$ in the search stage. For the $T = 50$ step configuration, the success rate is 68.7%. The FID value is 34.00, with lower values suggesting better image quality. The IS is reported as 93.94±1.84, with higher values indicating diverse and high-quality images. The TS value is 0.97, representing a high level of text similarity. Moving on to the $T = 100$ step configuration, the success rate increases to 81.2%, showing an improvement compared to the previous configuration. The FID value decreases to 29.65, indicating better image quality. The IS is reported as 66.09±1.83, showing a slight decrease compared to the previous configuration. The TS value is 0.84, suggesting a slight decrease in text similarity. In the $T = 150$ step configuration, the success rate decreases to 67.2%, slightly lower than the initial configuration. The FID value increases to 45.23, suggesting a decrease in image quality. The IS is reported as 58.51±0.79, indicating a decrease in the diversity and quality of generated images. The TS value remains relatively stable at 0.82.

When using $T = 50$, the attack prompt fails to fit well and exhibits a higher text similarity with the clean prompt. Although the generated images at this stage still maintain good quality and closely resemble those generated by the clean prompt, the success rate of the attack is very low. On the other hand, when $T = 150$, overfitting occurs, resulting in a decrease in text similarity and image quality due to the overfitted attack prompt. Consequently, the success rate of the attack also decreases. Overall, the configuration of $T = 100$ proves to be appropriate.

## C.4 The Impact of Constraints

Table 3 examines the impact of the fluency and semantic similarity (BERTScore) constraints. When no constraints are applied, the attack success rate is notably high at 91.3%. However, this absence of constraints results in a lower text similarity (TS) score of 0.37, indicating a decreased resemblance to clean text and a decrease in image quality. By introducing fluency constraints alone, the attack success rate decreases to 81.7% but increases the text similarity to 0.79. Furthermore, incorporating semantic similarity constraints independently also leads to a slight reduction in success rate to 89.8%, but only marginally improves the text similarity to 0.51. The introduction of constraints, particularly fluency constraints, leads to an increase in text similarity. The fluency constraint takes into account the preceding tokens of each token, enabling the integration of contextual information for better enhancement of text similarity. On the other hand, BERTScore considers a weighted sum, focusing more on the similarity between individual tokens without preserving the interrelation between context. In other words, the word order may undergo changes as a result and leads to a low text similarity. Certainly, this outcome was expected, as BERTScore itself prioritizes the semantic consistency between two prompts, while the order of context may not necessarily impact semantics. This further highlights the importance of employing both constraints simultaneously. When both constraints are utilized together, the text similarity is further enhanced to 0.84. Meanwhile, the success rate of the

Table 4: Results of the ablation study on the samplers

| Sampler | Success (%) | FID (↓) | IS (↑) | TS (↑) |
|---|---|---|---|---|
| DDIM [31] | 81.2 | 29.65 | 66.09±1.83 | 0.84 |
| DPM-Solver [17] | 76.5 | 27.23 | 81.31±2.09 | 0.88 |

Table 5: Classification Accuracy of Different Classifiers on both clean short and long text prompts.

| Classifiers | Short Text Prompts | Long Text Prompts |
|---|---|---|
| ViT-B/16(CLIP) | 82.0% | 78.8% |
| ViT-B/32(CLIP) | 80.8% | 74.1% |
| ViT-L/14(CLIP) | 84.0% | 80.8% |
| ResNet50(CLIP) | 77.2% | 69.9% |
| ResNet50 [10] | 10.7% | 10.8% |
| Swin-B [16] | 79.8% | 76.2% |
| ViT-B/16 [6] | 82.0% | 78.9% |

attack (81.2%) is comparable to that achieved when employing only the fluency constraint, while the text similarity surpasses that obtained through the independent usage of the two constraints.

## C.5 Different Samplers

Table 4 illustrates the effectiveness of our attack method in successfully targeting both DDIM and the stronger DPM-Solver. For the DDIM sampler, our attack method achieves a success rate of 81.2%, indicating its ability to generate successful attack prompts. Similarly, our attack method demonstrates promising results when applied to the DPM-Solver sampler. With a success rate of 76.5%, it effectively generates attack prompts. The TS scores of 0.84 and 0.88, respectively, indicate a reasonable level of text similarity between the attack prompts and clean prompts. These outcomes demonstrate the transferability of our attack method, showcasing its effectiveness against both DDIM and the more potent DPM-Solver sampler.

## C.6 Ensemble of Multiple Classifiers

The amalgamation of multiple models, often termed as "ensemble or reassembly," can significantly enhance model performance [35]. The efficacy of the attack is potentially contingent upon the performance of the classifiers. Notably, even when the stable diffusion model generated the desired subjects with precision, there remains a possibility that the classifier may erroneously classify it. To eliminate the impact of the performance of a single classifier, we design a voting mechanism by using an ensemble of multiple classifiers. To be specific, the attack is defined as a failure if at least one classifier in the ensemble still recognizes the original categories in the text prompt. When we took this mechanism to evaluate the generated image after our attack, the success rate of the attack on the long and short prompts remained the same, 81.8% and 91.1% respectively. This further proves the effectiveness of our algorithm. Additionally, we test the voting mechanism with images generated from clean prompts covering the 1000 classes in ImageNet. As shown in the Table 5, most of the classifiers have a relatively high Top1 accuracy. We conducted experiments with three distinct classifiers: CLIP+VIT-B/16, Swin-B, and the vanilla VIT-B/16. The classification accuracies of these models were evaluated using both long and short text prompts. For the CLIP+VIT-B/16 model, the classification accuracies with long and short text prompts were found to be 78.80% and 82%, respectively. The Swin-B model demonstrated classification accuracies of 76.20% with long text prompts and 79.80% with short ones. Lastly, the vanilla VIT-B/16 model achieved classification accuracies of 78.90% and 82% with long and short text prompts, respectively. The probability that at least one of them successfully recognizes the current generated image is 90.40% and 92.30% for long and short text prompts, respectively, which is much higher than using only one classifier. This proves that our mechanism is effective.

