# Stable Diffusion is Unstable

## (Supplementary Material)

In this supplementary material, we first present a review of related works (Section A), including the diffusion model and studies about the vulnerabilities within text-to-image models. Following that, we delve into additional analyses concerning the vulnerabilities observed in the Stable Diffusion model (Section B). Subsequently, we offer instances of long and short prompt attacks, accompanied by the corresponding generated images, as well as instances of black-box attacks (Section C). Lastly, we undertake a comprehensive series of experiments to substantiate the effectiveness of our approach (Section D). These experiments include the evaluation of attacks targeting both long and short prompts. Additionally, ablation studies are conducted to explore attacks employing different search steps, assess the influence of our fluency and semantic similarity constraints on text similarity, and target diverse samplers (e.g., DDIM and DPM-Solver) in the attack process.

## A  Related Work

### A.1  Diffusion Model.

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

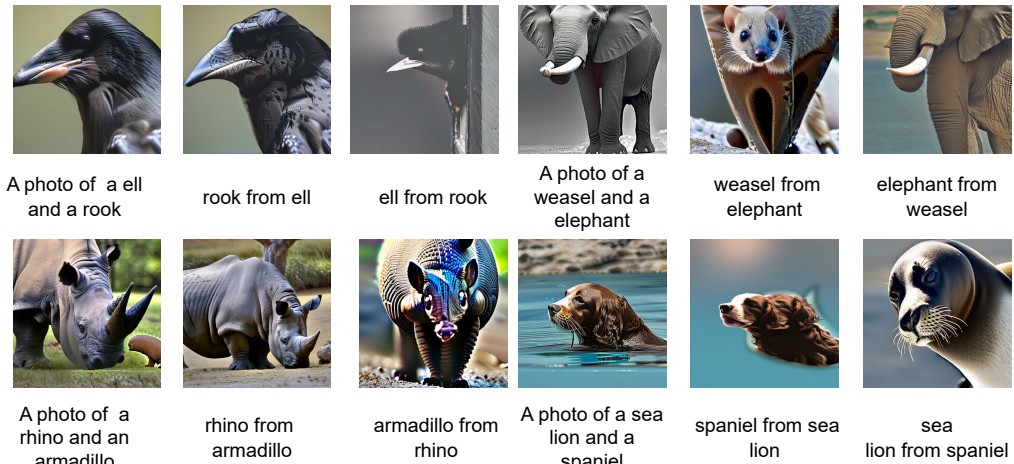

| A photo of a ell and a rook | rook from ell | ell from rook | A photo of a weasel and a elephant | weasel from elephant | elephant from weasel |

| A photo of a rhino and an armadillo | rhino from armadillo | armadillo from rhino | A photo of a sea lion and a spaniel | spaniel from sea lion | sea lion from spaniel |

Figure B.3: The image caption, "A photo of class$A$ and class$B$" represents the generated image when feature entanglement occurs; And "class$A$ from class$B$" represents the final generated image of prompt "A photo of class$A$" based on the eighth step of the prompt "A photo of class$B$"

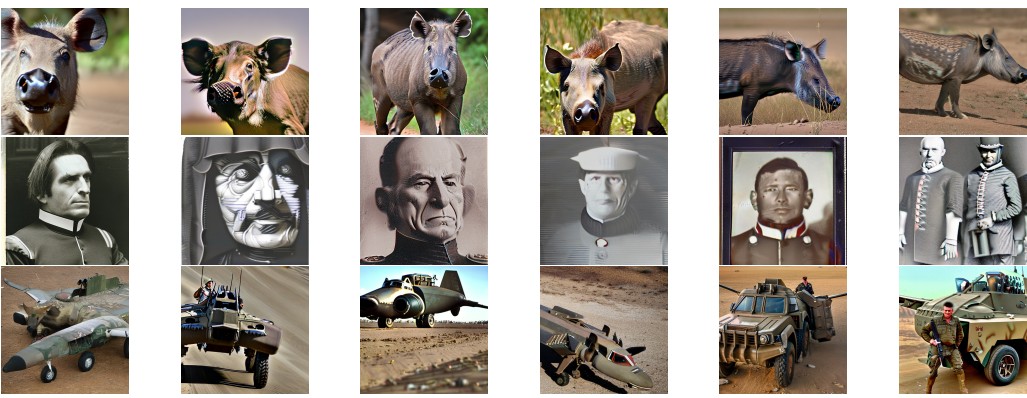

Figure B.4: The images in the first row are generated by the prompt "A photo of a warthog". The images in the second row are generated by the prompt "A photo of a traitor". The images in the third row are generated by the prompt "A photo of a warthog and a traitor".

minimum LPIPS values. In the concluding stages of generation, the distribution's median is found closer to the minimum LPIPS value, implying that the majority of classes are nearing completion. However, the sustained high maximum value suggests that there are classes still exhibiting slower generation rates.

## B.2 Pattern 2: Similarity of Coarse-grained Characteristics

To further verify that coarse-grained feature similarity is the root cause of feature entanglement, we provide more cases in Fig.B.3. From these cases, we can see that for the two classes where feature entanglement can occur, they can both continue the image generation task based on each other's coarse-grained information.