# OpenReview forum: "Stable Diffusion is Unstable"
_NeurIPS.cc/2023/Conference — NeurIPS 2023 spotlight_

### Official Review · Reviewer_L6BS · 2023-06-25

**Soundness:** 3 good
**Presentation:** 3 good
**Contribution:** 3 good
**Rating:** 6
**Confidence:** 4

**Summary:**

This paper finds some vulnerabilities of stable diffusion model, and proposes an auto-attack model to generate attack prompts.

**Strengths:**

This paper is well written and is easy to understand.
This paper discusses some vulnerabilities of the stable diffusion model, and does many experiments to verify.
The methodology illustration is clear and easy to follow.

**Weaknesses:**

The purpose and motivation of ATM is not well explained. Authors have find some vulnerabilities of the stable diffusion model, but the motivation of ATM is not correlated well with them. It would be better to connect the ATM and the aforementioned vulnerabilities.



**Questions:**

Are there some related work in this field? I would be better to mention some related work, and discuss or compare them in your experiments.

When describing the Variability in Generation Speed, the definition of speed is little confusing. What does the speed value represent? How to understand it?

**Limitations:**

This paper proposes an attack model, but lack of discussion of defense model. It would be better to discuss them together.

---

> ### Author Rebuttal · Authors · 2023-08-09
>
> We would like to thank the reviewers for their diligent review of this paper and for providing constructive feedback.
>
> **RE Purpose and Motivation:** Stable diffusion along with other text-to-image generative models (e.g., Midjourney and DALL-E 2) have wide-ranging applications and implications in the field of AIGC. Firstly, in practical application scenarios, the model may receive a variety of text prompts. Therefore, we need to explore under what form of text prompts, the text-to-image model will not be able to complete the image generation task, so as to improve the stability of the stable diffusion model. Our ATM algorithm is able to retrieve multiple attacking prompts based on a clean text prompt that prevents the stable diffusion model from generating images that match human aesthetics. Secondly, the current stable diffusion model still has many defects, some may be more common, but some are more invisible defects. Based on our algorithm to generate a large number of failure cases, researchers can systematically analyze the logic behind the failure cases. Also, the four vulnerabilities mentioned in the paper are summarized by observing these cases.
>
>
> **RE Discussion of Defense Model:** Our method can inspire the design of defense strategies and can work as an effective and reliable metric to evaluate those strategies. We design some possible defense strategies against the four patterns of vulnerability we found as follows:
>
> **Pattern 1. Variation in generation speed:** The speed at which the model generates images may depend on the relative generation speed in the text prompt, which may cause image features of one **noun** to overwrite those of another, or the sampling process enters into fine-grained generation information, the coarse-grained information of one of the **noun** has not been fully generated (the experiment that cannot generate categories without coarse-grained information can be seen in the third row of Figure 6 in the paper). defense mechanism may include:
>
> 1. The development of new training strategies, so that the model can generate image features of each **noun**  in a more balanced manner.
> 2. Adopt or design a new mechanism to try to solve the coordination problem in the sampling process of each token in the prompt. Intuitively, it is to keep the generation speed of each token consistent, so as to solve the problem of content disappearing in the generated images.
> 3. Add or design diversity losses or regularization terms for model training to encourage the model to generate more balanced and diverse images.
>
> **Pattern 2. The similarity of coarse-grained features:** When two **noun** have similar coarse-grained features, the model may generate images that mix the features of these two ** noun**. Defense mechanisms may include:
> 1. enhancing the feature discriminative ability of the model so that it can better handle **noun** that are different but have similar coarse-grained properties.
> 2. Introduce adversarial training to make the model better learn the differences between different **nouns**, instead of only focusing on their shared characteristics.
> 3. For coarse-grained similarity, we can preset multiple anchors during the generation process, forcing the category words in the prompt to be generated only in the corresponding anchors to solve the problem of attention map overlap.
>
> **Pattern 3. Polysemy of Words:** When **noun** is polysemy, the model may produce images that do not match user expectations. Defense mechanisms may include a more comprehensive consideration of the semantic context of **noun** during model training, enabling models to more accurately understand the specific meaning of **noun**. A solution might be to introduce context-aware language models to enhance the model's understanding of lexical polysemy.
>
> **Pattern 4. Word position:** The position of a descriptor in a sentence may affect the image generated by the model. Defense mechanisms may include
> 1. Train the model to better understand the location information of **noun** to reduce the impact of location changes on generated images.
> 2. Design a module that can manually adjust the weight of each word to eliminate the influence of location information.
>
>
> **RE Related Work:** In the section of A in Supplementary Material we refer to previous related work in exploring and addressing the stability of stable diffusion model generation as follows. DAAM performs a text-image attribution analysis on conditional text-to-image model and produces pixel-level attribution maps. Their research focuses on the phenomenon of feature entanglement and uncovers that the presence of cohyponyms may degrade the quality of generated images and that descriptive adjectives can attend too broadly across the image. Attend-and-Excite investigates the presence of catastrophic neglect in the Stable diffusion model, where the generative model fails to include one or more of the subjects specified in the input prompt. Additionally, they discover instances where the model fails to accurately associate attributes such as colors with their respective subjects. Although those works have some progress, there is still work to be done to enhance the stability and reliability of text-to-image models, ensuring consistent and satisfactory results for a wide range of text prompts， StructureDiffusion discovers that some attributes in the prompt not assigned correctly in the generated images, thus they employ consistency trees or scene graphs to enhance the embedding learning of the prompt.
>
> **RE Definition of Speed:** Take the generation process of an image as an example, stable diffusion requires N steps of the denoising process. We take the last step image as the reference image and use SSIM and LPIPS to calculate the distance from each step of the image to the last step. The speed of this distance converging to 1 (for SSIM) or 0 (for LPIPS) is the generation speed.

---

> > ### Comment · Reviewer_L6BS · 2023-08-14
> >
> > Thanks for your replies. Your response has solved my doubts very well. I would like to raise my score to acceptance.

---

### Official Review · Reviewer_czYq · 2023-06-29

**Soundness:** 3 good
**Presentation:** 3 good
**Contribution:** 3 good
**Rating:** 6
**Confidence:** 4

**Summary:**

The paper introduces Auto-attack on Text-to-image Models (ATM), a method to efficiently generate attack prompts that closely resemble clean prompts.
The method modifies text prompts by replacing or extending words, using a Gumbel Softmax distribution for differentiability.
It further applies a binary mask to preserve the desired object noun and imposes fluency and similarity constraints to ensure similarity.
The authors identify four distinct attack patterns through this attack method.
The method utilizes Stable Diffusion as the target model, allowing for white-box attacks, and demonstrates the transferability of these attacks to other generative models (black-box attacks).

**Strengths:**

1. The authors introduce a novel method to generate successful attack prompts in text-to-image generation pipelines. It can aid in vulnerability investigation. The method enables the identification of a broader range of attack patterns, prompting further research in both attack and defense mechanisms. Therefore, it can enhance security in the industry.

2. Related to the previous point, the authors themselves employ their method to uncover four distinct attack patterns, which are highly enlightening. This convincingly demonstrates the significant importance of this approach in studying the vulnerability of generative models and enhancing their robustness.

3. In their experiments, the proposed method demonstrates a high success rate in white-box attacks. It also exhibits excellent transferability to other generative models, e.g. DALL-E 2 and Midjourney. This proves that the method has a broad range of applications and can be used for various generative models, not just limited to Stable Diffusion.

**Weaknesses:**

1. Two types of modifications, namely replacing and extending, are considered. It can be confusing to determine when to use each type. How do they determine whether to replace or extend a word?

2. Most of the analyses are solid. However, in Section 3.1, the authors mention embedding two prompts c_1 and c_2 separately to "eliminate the additional impact of all possible extraneous factors". It is unclear what specific impact they want to eliminate and how embedding prompts separately can achieve this.

3. The authors mention the use of the method to design defensive strategies but do not discuss how to utilize it. Is it possible to provide an example?

4. A minor issue: In Fig. 6, the steps should range from 49 to 0 instead of 1 to 50. This is because in the reverse diffusion process, T represents noise, and 0 represents the generated image (similar to the steps shown in Fig. 4).

**Questions:**

Please address questions mentioned in "Weaknesses".

**Limitations:**

There are no potential negative societal impacts.

---

> ### Author Rebuttal · Authors · 2023-08-09
>
> We would like to thank the reviewers for their diligent review of this paper and for providing constructive feedback.
>
> **RE Replacing and Extending:** The two types of modifications can be automatically selected by our sampling mechanism and can be optimized using gradients. In the process of prompt tokenization, we uniformly pad all the prompts into the maximum length that the Clip can accept, which is 77.  After learning the attack distribution, when sampling by Gumble-Softmax, if the maximum value falls in the corresponding position of the clean text, it is the replacing, otherwise extending. For short text prompts, the prompt template (i.e., “A photo of a [NOUN]”) will affect the quality of the final generated image if the template is modified, some poor-quality images may be generated and make the classifier unable to recognize, but the category of the generated image is still the original category. In addition, the category keywords in the prompt also cannot be changed, because we want to remain the noun in the prompt and make the DM generate other objectives. Directly replace the original category keywords to make the original category disappear in the generated image, which is not a reasonable attack method. so only the extension is adopted.
>
> **RE Additional Impact:** In section B.4 in the supplementary material, we mentioned that the different positions of the category keywords in the prompt will lead to different contents in the final generated image. The reason is that there is a difference between the language model and humans in the understanding of the text. Even if the relative order of the category words is changed, the semantics will not change for humans, but for Clip’s text encoder, there will be a certain gap in the extracted semantic information. Thus, we first split one prompt into two parts and then concatenate the two embeddings, which can cut off the positional embedding of the two keywords, therefore eliminating the impact of the positional information.
>
> **RE Defensive Strategies:** Our algorithm can generate a large number and variety of attack text prompts, which is difficult to achieve manually.
>
> **(1)** Researchers can discover more vulnerabilities in the stable diffusion model by observing and analyzing these attack samples and designing an effective algorithm to improve the stability of the stable diffusion model based on the observed failure cases. We design some possible defense strategies against the four patterns of vulnerability we found as follows:
>
> **Pattern 1. Variation in generation speed:** The speed at which the model generates images may depend on the relative generation speed in the text prompt, which may cause image features of one **noun** to overwrite those of another, or the sampling process enters into fine-grained generation information, the coarse-grained information of one of the **noun** has not been fully generated (the experiment that cannot generate categories without coarse-grained information can be seen in the third row of Figure 6 in the paper). defense mechanism may include:
>
> 1. The development of new training strategies so that the model can generate image features of each **noun**  in a more balanced manner.
> 2. Adopt or design a new mechanism to try to solve the coordination problem in the sampling process of each token in the prompt. Intuitively, it is to keep the generation speed of each token consistent, so as to solve the problem of content disappearing in the generated images.
> 3. Add or design diversity losses or regularization terms for model training to encourage the model to generate more balanced and diverse images.
>
> **Pattern 2. The similarity of coarse-grained features:** When two **noun** have similar coarse-grained features, the model may generate images that mix the features of these two **noun**. Defense mechanisms may include:
> 1. enhancing the feature discriminative ability of the model so that it can better handle **noun** that are different but have similar coarse-grained properties.
> 2. Introduce adversarial training to make the model better learn the differences between different **nouns**, instead of only focusing on their shared characteristics.
> 3. For coarse-grained similarity, we can preset multiple anchors during the generation process, forcing the category words in the prompt to be generated only in the corresponding anchors to solve the problem of attention map overlap.
>
> **Pattern 3. Polysemy of Words:** When **noun** is ambiguous, the model may produce images that do not match user expectations. Defense mechanisms may include a more comprehensive consideration of the semantic context of **noun** during model training, enabling models to more accurately understand the specific meaning of **noun**. A solution might be to introduce context-aware language models to enhance the model's understanding of lexical polysemy.
>
> **Pattern 4. Word position:** The position of a descriptor in a sentence may affect the image generated by the model. Defense mechanisms may include
> 1. Train the model to better understand the location information of **noun** to reduce the impact of location changes on generated images.
> 2. Design a module that can manually adjust the weight of each word to eliminate the influence of location information.
>
>
> **(2)** We can use this algorithm to generate a large number of offensive text prompts and use these attack samples to create a data set specifically for generation stability. This data set can be used to test the stability of the diffusion generation model, and can also be used for Adversarial training, to improve the stability of the model.
>
> **RE minor issue:** Thank you for pointing out this mistake, we will correct it immediately.

---

> > ### Comment · Reviewer_czYq · 2023-08-15
> > **post-rebuttal comments**
> >
> > The authors have addressed my concerns.
> > It is a good paper.
> > I vote for acceptance.

---

### Official Review · Reviewer_adct · 2023-07-03

**Soundness:** 3 good
**Presentation:** 4 excellent
**Contribution:** 4 excellent
**Rating:** 7
**Confidence:** 4

**Summary:**

This paper proposes an adversarial attack against text-to-image models that can generate adversarial prompts to prevent the stable diffusion models from generating the desired subjects. The attack is gradient-based by utilizing the Gumbel Softmax to make the work embedding differentiable. Then, the authors provide a comprehensive analysis of the vulnerabilities of the stable diffusion models.

**Strengths:**


1. This paper is well-written and well-organized.

2. The proposed automatic attack is effective and convenient for implementation.

3. The authors provide interesting and inspiring analyses of the vulnerabilities of the generative models. They found the differences in the generation speed, the similarity of coarse-grained subjects in the prompt, and the polysemy are the important factors in the robustness of the generative models. This analysis is definitely beneficial to the future study of the robustness of generative models.

4. The authors show that the proposed attack is transferable as well.


**Weaknesses:**

1. Although the constraint in the attack seems to be novel, the attack objective is similar to the previous proposed C&W attacks.



**Questions:**

1. What are the computation consumption and requirement to conduct the white-box attack against the stable diffusion?

2. The attack success rate could be affected by the performance of the network. For example, the network could misclassify a subject even if the subject is correctly generated by the diffusion model. Could you discuss how to mitigate this issue during the evaluation procedure?

3. It seems the attack will make the FID score higher and IS score lower. Does it indicate a drawback of the attack? I am confused about whether we need to maintain the FID and IS score during the attack. Could the authors provide some explanations?


**Limitations:**

None.

---

> ### Author Rebuttal · Authors · 2023-08-09
>
> We would like to thank the reviewers for their diligent review of this paper and for providing constructive feedback.
>
> **RE Objective and C&W:**
> Our objective differs from that of C&W from three perspectives:
>
> 1. **The type of target models:** C&W attacks commonly target classification models, aiming to alter the model's classification results. On the other hand, in our optimization objective, a generative model is targeted, aiming to alter the content of the generated image to make it different from the target class. Although our pipeline also includes a classification model, its role is to measure the success of the attack, and this classification model is not our target of attack.
> 2. **The continuity/discreteness and differentiability of perturbations:** In C&W attacks, perturbations are applied to image inputs, which are continuous and differentiable (within the range of 0.0 to 1.0). In our optimization objective, the inputs consist of words represented by a dictionary of embedding vectors, which are discrete and non-differentiable. Therefore, it is necessary to introduce a mechanism that renders them continuous and differentiable, thereby enabling gradient-based optimization. To be specific, Gumbel Softmax distributions are introduced to make the perturbations continuous and differentiable. Two types of modifications, namely replacing and extending are to automatically learn the replacement or expansion of words, and then pass a text-to-image transformation function (i.e., stable diffusion) to detect the effectiveness of the perturbation.
> 3. **The complexity of constraints:** The constraints applied in C&W attacks include regularization terms to restrict the p-norm of perturbations and the clamping of pixel values within the range of 0.0 to 1.0 after the attack. These terms are used to make perturbations very small and thus difficult for humans to detect. However, in our method, p-norm and clamping cannot be used since the difference between the text and image. Therefore, we design constraints that are more suitable for the textual data,i.e., fluency constraints, and BERT similarity constraints. This constraint not only limits the size of the perturbation, but also limits how similar the perturbed text is to the original text, and how close the generated distribution corresponding to the perturbation is to the true distribution.
>
> **RE Computation Consumption:** Our algorithm consumes 0.1 GPU Hours to learn the attack distribution of a sample on a single RTX3090 with 100 steps of gradient search, and 0.075 GPU Hours to sample 100 attack prompts from this distribution, so for a single sample, it consumes a total of 0.175 GPU Hours
>
> **RE Attack Success Rate:** The classifier we use for the attack is CLIP ViT-B/16.
> For eliminate the impact of the performance of a single classifier, we design a voting mechanism by using an ensemble of multiple classifiers. To be specific, the attack is defined as a failure if at least one classifier in the ensemble still recognizes the original categories in the text prompt. When we took this mechanism to evaluate the generated image after our attack, the success rate of the attack on the long and short prompts remained the same, 81.8% and 91.1% respectively. This further proves the effectiveness of our algorithm. Additionally, we test the voting mechanism with images generated from clean prompts covering the 1000 classes in ImageNet. As shown in the PDF, most of the classifiers have a relatively high Top1 accuracy. We conducted experiments with three distinct classifiers: CLIP+ViT-B/16, Swin-B, and the vanilla ViT-B/16. The classification accuracies of these models were evaluated using both long and short text prompts. For the CLIP+ViT-B/16 model, the classification accuracies with long and short text prompts were found to be 78.80% and 82%, respectively. The Swin-B model demonstrated classification accuracies of 76.20% with long text prompts and 79.80% with short ones. Lastly, the vanilla ViT-B/16 model achieved classification accuracies of 78.90% and 82% with long and short text prompts, respectively. The probability that at least one of them successfully recognizes the current generated image is 90.40% and 92.30% for long and short text prompts, respectively, which is much higher than using only one classifier. This proves that our mechanism is effective.
>
> **RE FID and IS Scores:** This is not a drawback, instead this is an echo of our algorithm design. The attack objective of our algorithm, namely the margin loss, is used to reduce the classifier’s confidence in the true class $y$ and improve its confidence in the class with the largest confidence excluding $y$, until a margin $\kappa$ is reached. Thereby suppressing the categories in the clean text prompt from appearing in the generated images. Therefore, from the perspective of this loss function, whether it is to reduce the quality of the image or make the original category disappear from the generated image, both align with this expectation.
>
> FID is a metric that measures the distance between the distribution of generated images and the reference dataset. In our paper, the reasons for the FID metrics increased can be concluded as follows. Firstly, we use the ImageNet validation set as the reference dataset, the content of the image generated after the attack will deviate from the distribution of this reference dataset, as the nouns used for replacement or expansion might not exist within ImageNet's 1000 categories.
>
> The second point is that after the attack, images generated by the stable diffusion model indeed contain low-quality images. This allows us to explore the types of prompts that can diminish the ability to generate images of stable diffusion, which is also one of our attack purposes. Those aforementioned images will not only increase the FID but also reduce the IS score.

---

> > ### Comment · Reviewer_adct · 2023-08-14
> > **Solved my concerns**
> >
> > Thanks for your replies. I appreciate that the authors used a voting mechanism to eliminate the impact of the performance of a single classifier, which definitely makes the attack success rate more convincible. Other concerns are well solved as well. Therefore, I still lean towards Acceptance.

---

### Official Review · Reviewer_kKNr · 2023-07-06

**Soundness:** 3 good
**Presentation:** 3 good
**Contribution:** 3 good
**Rating:** 7
**Confidence:** 4

**Summary:**

In this paper, the authors use Gumble softmax as well as the gradient based method to learn the distribution of an attack text prompt, defined as an attack text prompt that enables a text-to-image generation model to generate images that do not match the text description without changing the category keywords in the clean text prompt, for exploring the vulnerability of stable diffusion model. Based on the failure cases generated by the algorithm, the authors place the possible reasons for the failure of the generation into four main categories.

**Strengths:**

1. This algorithm able to automatically finds the prompt that causes the text to image generation model to fail, offering the possibility to explore the vulnerability of the stable diffusion model systematically.

2. Various patterns of generative failure have been identified based on a large number of generative failure cases, and a number of reasonable experiments have been designed to further support these conjectures.

3. This algorithm able to learn multiple attack text prompts based on a clean prompt, further boosting the number of generated failure cases.

**Weaknesses:**

1. It seems that in pattern 1 only the speed of generation of different categories under the same random seed was explored, would the speed of generation of the same category with different random seeds also be different?

2. In the section on quantitative experiments, what is the classification accuracy of CLIP for clean text prompts?

3. How do you handle the gradient of 50 steps in the stable diffusion model?

4. Most contents in this paper are clear and detailed, but the setting in the quantitative experiments section lacks some details, e.g. it does not explain how the random algorithm changed the text prompt, which Casual Language Model was used, and which version of Clip was used.

**Questions:**

See weakness

---

> ### Author Rebuttal · Authors · 2023-08-09
>
> We would like to thank the reviewers for their diligent review of this paper and for providing constructive feedback.
>
> **RE Generation Speed with Different Random Seeds:** We prefer to focus on the relative generation speed difference of different categories under the same initial noise since the different categories in the same prompt share the same initial noise. As shown in the pdf, we selected "tench" as an example, and used 100 different initial noises to generate 100 images of tench, and the results show that even for the same category, there is a significant difference in the generation speed under different random seeds. Although this phenomenon exists, it does not affect our previous conclusion that there is a difference in generation speed between different classes with the same initial noise
>
> **RE Classification Accuracy:** In the process of learning attack prompt distribution and the sampling of Gumble-Softmax, we have used the pre-trained ViT-B/16 Clip as the classifier, which has a classification error of 18% in the short text, and when perturbed using a randomized algorithm, the classification error rate rises from 18% to 79.2%, and our proposed algorithm improves it from 18% to 91.1%. In the long text prompt, Clip's probability of classification error is 21.2%, and with the randomized algorithm, the classification error rate is only improved to 41.4%, while our algorithm improves it to 81.2%, and based on these results, the effectiveness of our algorithm is proved.
>
> Additionally, we also add an experiment with an ensemble of various classifiers, which provides an improved classification performance. To be specific, the attack is defined as a failure if at least one of them still recognizes the original categories of this text prompt. This approach may eliminate the impact of the performance of a single classifier. When we took this mechanism to evaluate the generated image after our attack, the success rate of the attack on the long and short prompts remained the same, 81.8% and 91.1% respectively. thus further proving the effectiveness of our algorithm. To this end, we test the voting mechanism with images from the  1000 classes in ImageNet generated by the stable diffusion model. As shown in the table below, most of the classifiers have a relatively high Top1 accuracy.  We conducted experiments with three distinct classifiers: CLIP+ViT-B/16, Swin-B, and the vanilla ViT-B/16. The classification accuracies of these models were evaluated using both long and short text prompts. For the CLIP+ViT-B/16 model, the classification accuracies with long and short text prompts were found to be 78.80% and 82%, respectively. The Swin-B model demonstrated classification accuracies of 76.20% with long text prompts and 79.80% with short ones. Lastly, the vanilla ViT-B/16 model achieved classification accuracies of 78.90% and 82% with long and short text prompts, respectively. The probability that at least one of them successfully recognizes the current generated image is 90.40% and 92.30% for long and short text prompts, respectively, which is much higher than using only one classifier. This proves that our mechanism is effective.
>
>
> |  | Short Text Promot | Long Text Prompot |
> |-----|:-----:|:-----:|
> | ViT-B/16(CLIP) | 82.0% | 78.8% |
> | ViT-B/32(CLIP) | 80.8% | 74.1% |
> | ViT-L/14(CLIP) | 84.0% | 80.8% |
> |ResNet50(CLIP)| 77.2% | 69.9% |
> | ResNet50 | 10.7% | 10.8% |
> | Swin-B | 79.8% | 76.2% |
> | ViT-B/16 | 82.0% | 78.9% |
>
> **RE Gradient:** In order to save GPU computing cost and memory, in the 50-step sampling process, we turn off the gradient of image and text in the first 49 steps and only keep the gradient of the last step. According to our quantitative experimental results, our algorithm can effectively improve the classification error rate of the CLIP classifier for both short text and long text.
>
> **RE Lack of Details:**
> 1. The random algorithm extends or replaces words randomly.  For the long text prompts, we replace one or two words at random positions (without category keywords) and extend one word at the end. For short text prompts, the prompt template (i.e., “A photo of a [NOUN]”) will affect the quality of the final generated image, if the template is modified, some poor-quality images may be generated and make the classifier unable to recognize, but the category of the generated image is still the original category. In addition, the category keywords in the prompt also cannot be changed, because we want to remain the noun in the prompt and make the DM generate other objectives. Directly replace the original category keywords to make the original category disappear in the generated image, which is not a reasonable attack method. Therefore, we only extend one or two words at the end of the clean prompt, these words are randomly selected from the vocabulary of CLIP.
> 2. CLM is GPT2. It is trained from scratch with the wikitext-103 L dataset under Clip's tokenizer. Because in our optimization goal, we use GP2 as the reference model to constrain the language fluency of the generated attack text prompt. For the same text, different tokenizers will map different input ids. For the language model, That is, different semantics, and there is currently no open source GPT2 trained by a Clip tokenizer, so we need to retrain a GPT2 based on a Clip tokenizer.
> 3. The version of  Clip is ViT-B/16.

---

### Author Rebuttal · Authors · 2023-08-09

We would like to thank the reviewers for their diligent review of this paper and for providing constructive feedback.

This PDF includes a Violin plot illustrating the generation speed of the same class with different initial noises. Additionally, there is a table that details the classification accuracy of various classifiers on both clean long and short text prompts.

---

### Comment · Area_Chair_W7kP · 2023-08-19
**Discussions are required for the submission**

Dear all reviewers，

Thank you very much for your great efforts in reviewing the referred submission. Now the authors have provided responses regarding your concerns. Would you please read the authors response and see whether your concerns have been addressed or not. You are welcome to raise further concerns if necessary so that the authors can respond to them timely. Your great service would be very important for the community to make final decisions.

Best regards，
Your AC

---

### Decision · Program_Chairs · 2023-09-21

**Decision:**

Accept (spotlight)

**Comment:**

This paper proposes an adversarial attack against text-to-image models that can generate adversarial prompts to prevent the stable diffusion models from generating the desired subjects. The attack is gradient-based by using Gumbel Softmax to make the work embedding differentiable. The authors provide a comprehensive analysis of the vulnerabilities of the stable diffusion models. The proposed method enables the identification of a broader range of attack patterns, prompting further research in both attack and defense mechanisms, which would be useful to enhance security in the industry.